# THE UNREASONABLE EFFECTIVENESS OF RANDOM PRUNING: RETURN OF THE MOST NAIVE BASELINE FOR SPARSE TRAINING

**Shiwei Liu[1], Tianlong Chen[2], Xiaohan Chen[2], Li Shen[3]**
**Decebal Constantin Mocanu[1,4], Zhangyang Wang[2], Mykola Pechenizkiy[1]**,
[1]Eindhoven University of Technology, [2]University of Texas at Austin
[3]JD Explore Academy,[4]University of Twente,
{s.liu3,m.pechenizkiy}@tue.nl,
{tianlong.chen,xiaohan.chen,atlaswang}@utexas.edu,
d.c.mocanu@utwente.nl,mathshenli@gmail.com

## ABSTRACT

Random pruning is arguably the most naive way to attain sparsity in neural networks, but has been deemed uncompetitive by either post-training pruning or sparse training. In this paper, we focus on sparse training and highlight a perhaps counter-intuitive finding, that random pruning at initialization can be quite powerful for the sparse training of modern neural networks. Without any delicate pruning criteria or carefully pursued sparsity structures, we empirically demonstrate that sparsely training a randomly pruned network from scratch can match the performance of its dense equivalent. There are two key factors that contribute to this revival: (i) *the network sizes matter*: as the original dense networks grow wider and deeper, the performance of training a randomly pruned sparse network will quickly grow to matching that of its dense equivalent, even at high sparsity ratios; (ii) *appropriate layer-wise sparsity ratios* can be pre-chosen for sparse training, which shows to be another important performance booster. Simple as it looks, a randomly pruned subnetwork of Wide ResNet-50 can be sparsely trained to outperforming a dense Wide ResNet-50, on ImageNet. We also observed such randomly pruned networks outperform dense counterparts in other favorable aspects, such as out-of-distribution detection, uncertainty estimation, and adversarial robustness. Overall, our results strongly suggest there is larger-than-expected room for sparse training at scale, and the benefits of sparsity might be more universal beyond carefully designed pruning. Our source code can be found at https://github.com/VITA-Group/Random_Pruning.

## 1 INTRODUCTION

Most recent breakthroughs in deep learning are fairly achieved with the increased complexity of over-parameterized networks (Brown et al., 2020; Raffel et al., 2020; Dosovitskiy et al., 2021; Fedus et al., 2021. arXiv:2101.03961; Jumper et al., 2021; Berner et al., 2019). It is well-known that large models train better (Neyshabur et al., 2019; Novak et al., 2018; Allen-Zhu et al., 2019), generalize better (Hendrycks & Dietterich, 2019; Xie & Yuille, 2020; Zhao et al., 2018), and transfer better (Chen et al., 2020b;a; 2021b). However, the upsurge of large models exacerbates the gap between research and practice since many real-life applications demand compact and efficient networks.

Neural network pruning, since proposed by (Mozer & Smolensky, 1989; Janowsky, 1989), has evolved as the most common technique in literature to reduce the computational and memory requirements of neural networks. Over the past few years, numerous pruning criteria have been proposed, including magnitude (Mozer & Smolensky, 1989; Han et al., 2015; Frankle & Carbin, 2019; Mocanu et al., 2018), Hessian (LeCun et al., 1990; Hassibi & Stork, 1993), mutual information (Dai et al., 2018), Taylor expansion (Molchanov et al., 2016), movement (Sanh et al., 2020), connection sensitivity (Lee et al., 2019), etc. Motivated for different scenarios, pruning can occur after training (Han et al., 2015; Frankle & Carbin, 2019; Molchanov et al., 2016; Lee et al., 2021), during training (Zhu & Gupta,

2017; Gale et al., 2019; Louizos et al., 2018; You et al., 2020; Chen et al., 2021c;a), and even before training (Mocanu et al., 2018; Lee et al., 2019; Gale et al., 2019; Wang et al., 2020; Tanaka et al., 2020). The last regime can be further categorized into "static sparse training" (Mocanu et al., 2016; Gale et al., 2019; Lee et al., 2019; Wang et al., 2020) and "dynamic sparse training" (Mocanu et al., 2018; Bellec et al., 2018; Evci et al., 2020a; Liu et al., 2021b;a).

While random pruning is a universal method that can happen at any stage of training, training a randomly pruned network from scratch is arguably the most appealing way, owing to its "end-to-end" saving potential for the entire training process besides the inference. Due to this reason, we focus on random pruning for sparse training in this paper. When new pruning approaches bloom, random pruning naturally becomes their performance's empirical "lower bound" since the connections are randomly chosen without any good reasoning. Likely due to the same reason, the results of sparse training with random pruning (as "easy to beat" baselines to support fancier new pruning methods) reported in the literature are unfortunately vague, often inconsistent, and sometimes casual. For instance, it is found in Liu et al. (2020b) that randomly pruned sparse networks can be trained from scratch to match the full accuracy of dense networks with only 20% parameters, whereas around 80% parameters are required to do so in Frankle et al. (2021). The differences may arise from architecture choices, training recipes, distribution hyperparameters/layer-wise ratios, and so on.

In most pruning literature (Gale et al., 2019; Lee et al., 2019; Frankle et al., 2021; Tanaka et al., 2020), random pruning usually refers to randomly removing the same proportion of parameters per layer, ending up with uniform layer-wise sparsities. Nevertheless, researchers have explored other pre-defined layer-wise sparsities, e.g., uniform+ (Gale et al., 2019), *Erdős-Rényi* random graph (ER) (Mocanu et al., 2018), and *Erdős-Rényi-Kernel* (ERK) (Evci et al., 2020a). These layer-wise sparsities also fit the category of random pruning, as they require no training to obtain the corresponding sparsity ratios. We assess random pruning for sparse training with these layer-wise sparsity ratios, in terms of various perspectives besides the predictive accuracy.

Our main findings during this course of study are summarized below:

- We find that the network size matters for the effectiveness of sparse training with random pruning. With small networks, random pruning hardly matches the full accuracy even at mild sparsities (10%, 20%). However, as the networks grow wider and deeper, the performance of training a randomly pruned sparse network will quickly grow to matching that of its dense equivalent, even at high sparsity ratios.

- We further identify that appropriate layer-wise sparsity ratios can be an important booster for training a randomly pruned network from scratch, particularly for large networks. We investigate several options to pre-define layer-wise sparsity ratios before any training; one of them is able to push the performance of a completely random sparse Wide ResNet-50 over the densely trained Wide ResNet-50 on ImageNet.

- We systematically assess the performance of sparse training with random pruning, and observe surprisingly good accuracy and robustness. The accuracy achieved by ERK ratio can even surpass the ones learned by complex criteria, e.g., SNIP or GraSP. In addition, randomly pruned and sparsely trained networks are found to outperform conventional dense networks in other favorable aspects, such as out-of-distribution (OoD) detection, adversarial robustness, and uncertainty estimation.

## 2 RELATED WORK

### 2.1 STATIC SPARSE TRAINING

Static sparse training represents a class of methods that aim to train a sparse subnetwork with a fixed sparse connectivity pattern during the course of training. We divide the static sparse training into random pruning and non-random pruning according to whether the connection is randomly selected.

**Random Pruning.** Static sparse training with random pruning samples masks within each layer in a random fashion based on pre-defined layer-wise sparsities. The most naive approach is pruning each layer uniformly with the same pruning ratio, i.e., uniform pruning (Mariet & Sra, 2016; He et al., 2017; Suau et al., 2019; Gale et al., 2019). Mocanu et al. (2016) proposed a non-uniform and scale-free topology, showing better performance than the dense counterpart when applied to restricted

Boltzmann machines (RBMs). Later, expander graphs were introduced to build sparse CNNs and showed comparable performance against the corresponding dense CNNs (Prabhu et al., 2018; Kepner & Robinett, 2019). While not initially designed for static sparse training, ER (Mocanu et al., 2018) and ERK (Evci et al., 2020a) are two advanced layer-wise sparsities introduced from the field of graph theory with strong results.

**Non-Random Pruning.** Instead of pre-choosing a sparsity ratio for each layer, many works utilize the proposed saliency criteria to learn the layer-wise sparsity ratios before training, also termed as pruning at initialization (PaI). Lee et al. (2019) first introduced SNIP that chooses structurally important connections at initialization via the proposed connection sensitivity. Following SNIP, many efficient criteria have been proposed to improve the performance of non-random pruning at initialization, including but not limited to gradient flow (GraSP; Wang et al. (2020)), synaptic strengths (SynFlow; Tanaka et al. (2020)), neural tangent kernel (Liu & Zenke, 2020), and iterative SNIP (de Jorge et al., 2021; Verdenius et al., 2020). Su et al. (2020); Frankle et al. (2021) uncovered that the existing PaI methods hardly exploit any information from the training data and are very robust to mask shuffling, whereas magnitude pruning after training learns both, reflecting a broader challenge inherent to pruning at initialization.

## 2.2 DYNAMIC SPARSE TRAINING

In contrast to static sparse training, dynamic sparse training stems from randomly initialized sparse subnetworks, and meanwhile dynamically explores new sparse connectivity during training. Dynamic sparse training starts from Sparse Evolutionary Training (SET) (Mocanu et al., 2018; Liu et al., 2020a) which initializes the sparse connectivity with *Erdős-Rényi* (Erdős & Rényi, 1959) topology and periodically explores the sparse connectivity via a prune-and-grow scheme during the course of training. While there exist numerous pruning criteria in the literature, simple magnitude pruning typically performs well in the field of dynamic sparse training. On the other hand, the criteria used to grow weights back differs from method to method, including randomness (Mocanu et al., 2018; Mostafa & Wang, 2019), momentum (Dettmers & Zettlemoyer, 2019), gradient (Evci et al., 2020a; Jayakumar et al., 2020; Liu et al., 2021b). Besides the prune-and-grow scheme, layer-wise sparsities are vital to achieving high accuracy. (Mostafa & Wang, 2019; Dettmers & Zettlemoyer, 2019) reallocates weights across layers during training based on reasonable heuristics, demonstrating performance improvement. Evci et al. (2020a) extended ER to CNNs and showed considerable performance gains to sparse CNN training with the *Erdős-Rényi-Kernel* (ERK) ratio. Very recently, Liu et al. (2021a) started from a subnetwork at a smaller sparsity and gradually pruned it the target sparsity during training. The denser initial subnetwork provides a larger exploration space for DST at the early training phase and thus leads to a performance improvement, especially for extreme sparsities. Even though dynamic sparse training achieves promising sparse training performance, it changes the sparse connectivity during training and is thus out of the scope of random pruning.

While prior works have observed that random pruning can be more competitive in certain cases (Mocanu et al., 2018; Liu et al., 2020b; Su et al., 2020; Frankle et al., 2021), they did not give principled guidelines on when and how it can become that good; nor do they show it can match the performance of dense networks on ImageNet. Standing on the shoulders of those giants, our work summarized principles by more extensive and rigorous studies, and demonstrate the strongest result so far, that randomly pruned sparse Wide ResNet-50 can be sparsely trained to outperform a dense Wide ResNet-50, on ImageNet. Moreover, compared with the ad-hoc sparsity ratios used in (Su et al., 2020), we show that ERK (Evci et al., 2020a) and our modified ERK+ are more generally applicable sparsity ratios that consistently demonstrate competitive performance without careful layer-wise sparsity design for every architecture. Specifically, ERK+ ratio achieves similar accuracy with the dense Wide ResNet-50 on ImageNet while being data free, feedforward free, and dense initialization free.

## 3 METHODOLOGY

We conduct extensive experiments to systematically evaluate the performance of random pruning. The experimental settings are described below.

### 3.1 LAYER-WISE SPARSITIES RATIOS

Denote $s^l$ as the sparsity of layer $l$. Random pruning, namely, removes weights or filters in each layer randomly to the target sparsity $s^l$. Different from works that learn layer-wise sparsities and model

weights together during training using iterative global pruning techniques (Frankle & Carbin, 2019), soft threshold (Kusupati et al., 2020), and dynamic reparameterization (Mostafa & Wang, 2019), etc., the layer-wise sparsities of random pruning is pre-defined before training. Many pre-defined layer-wise sparsity ratios in the literature are suited for random pruning, while they may not initially be designed for random pruning. We choose the following 6 layer-wise sparsity ratios to study. SNIP ratio and GraSP ratio are two layer-wise ratios that we borrow from PaI.

**ERK.** Introduced by Mocanu et al. (2018), Erdős-Rényi (ER) sparsifies the Multilayer Perceptron (MLP) with a random topology in which larger layers are allocated with higher sparsity then smaller layers. Evci et al. (2020a) further proposed a convolutional variant (Erdős-Rényi-Kernel (ERK)) that takes the convolutional kernel into consideration. Specifically, the sparsity of the convolutional layer is scaled proportional to $1 - \frac{n^{l-1}+n^l+w^l+h^l}{n^{l-1} \times n^l \times w^l \times h^l}$ where $n^l$ refers to the number of neurons/channels in layer $l$; $w^l$ and $h^l$ are the corresponding width and height.

**ERK+.** We modify ERK by forcing the last fully-connected layer as dense if it is not, while keeping the overall parameter count the same. Doing so improves the test accuracy of Wide ResNet-50 on ImageNet[1] as shown in Appendix F.

**Uniform.** Each layer is pruned with the same pruning rate so that the pruned network ends up with a uniform sparsity distribution, e.g., Zhu & Gupta (2017).

**Uniform+.** Instead of using a completely uniform sparsity ratio, Gale et al. (2019) keep the first convolutional layer dense and maintain at least 20% parameters in the last fully-connected layer.

**SNIP ratio.** SNIP is a PaI method that selects important weights based on the connection sensitivity score $|g \odot w|$, where $w$ and $g$ is the network weight and gradient, respectively. The weights with the lowest scores in *one iteration* are pruned before training. While not initially designed for random pruning, we adjust SNIP for random pruning by only keeping its layer-wise sparsity ratios, while discarding its mask positions. New mask positions (non-zero elements) are re-sampled through a uniform distribution $\sim \text{Uniform}(0, 1)$ with a probability of $1 - s^l$. Such layer-wise sparsities are obtained in a (slightly) data-driven way, yet very efficiently *before training* without any weight update. The SNIP ratio is then treated as another pre-defined (i.e., before the training starts) sampling ratio for random pruning.

**GraSP ratio.** GraSP is another state-of-the-art method seeking to pruning at initialization. Specifically, GraSP removes weights that have the least effect on the gradient norm based on the score of $-w \odot Hg$, where $H$ is the Hessian matrix and $g$ is the gradient. Same with SNIP, we keep the layer-wise sparsity ratios of GraSP and re-sample its mask positions.

### 3.2 EXPERIMENTAL SETTINGS

Table 1: Summary of the architectures, datasets and hyperparameters used in this paper.

| Model | Mode | Data | #Epoch | Batch Size | LR | Momentum | LR Decay, Epoch | Weight Decay |
|---|---|---|---|---|---|---|---|---|
| ResNets | Dense | CIFAR-10/100 | 160 | 128 | 0.1 | 0.9 | 10×, [80, 120] | 0.0005 |
| | Sparse | CIFAR-10/100 | 160 | 128 | 0.1 | 0.9 | 10×, [80, 120] | 0.0005 |
| Wide ResNets | Dense | ImageNet | 90 | 192*4 | 0.4 | 0.9 | 10×, [30, 60, 80] | 0.0001 |
| | Sparse | ImageNet | 100 | 192*4 | 0.4 | 0.9 | 10×, [30, 60, 90] | 0.0001 |

**Architectures and Datasets.** Our main experiments are conducted with the CIFAR version of ResNet (He et al., 2016) with varying depths and widths on CIFAR-10/100 (Krizhevsky et al., 2009), the batch normalization version of VGG (Simonyan & Zisserman, 2014) with varying depths on CIFAR-10/100, and the ImageNet version of ResNet and Wide ResNet-50 (Zagoruyko & Komodakis, 2016) on ImageNet (Deng et al., 2009). For ImageNet, we follow the common setting in sparse training (Dettmers & Zettlemoyer, 2019; Evci et al., 2020b; Liu et al., 2021b) and train sparse models for 100 epochs. All models are trained with stochastic gradient descent (SGD) with momentum. We share the summary of the architectures, datasets, and hyperparameters in Table 1.

**Measurement Metrics.** In most pruning literature, test accuracy is the core quality that researchers consider. However, the evaluation of other perspectives is also important for academia and industry

---

[1]For datasets with fewer classes, e.g, MNIST, CIFAR-10, ERK+ scales back to ERK as the last fully-connected layer is already dense.

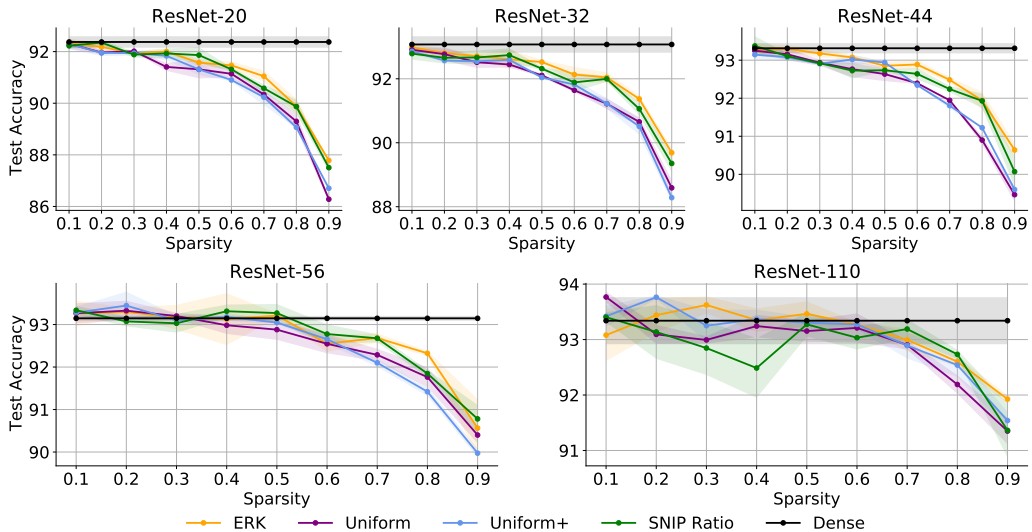

Figure 1: **From shallow to deep.** Test accuracy of training randomly pruned subnetworks from scratch with different depth on CIFAR-10. ResNet-A refers to a ResNet model with A layers in total.

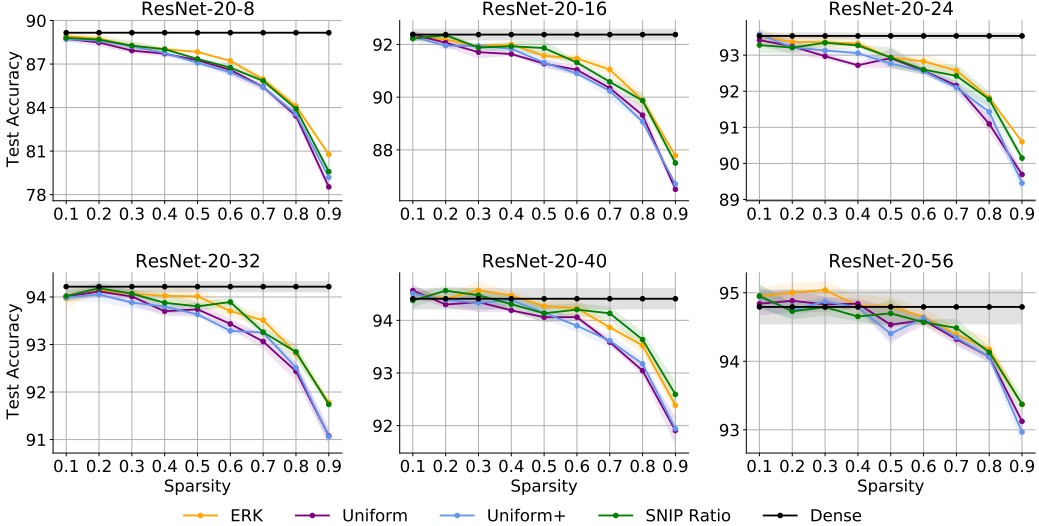

Figure 2: **From narrow to wide.** Test accuracy of training randomly pruned subnetworks from scratch with different width on CIFAR-10. ResNet-A-B refers to a ResNet model with A layers in total and B filters in the first convolutional layer.

before replacing dense networks with sparse networks on a large scope. Therefore, we thoroughly evaluate sparse training with random pruning from a broader perspective. Specifically, we assess random pruning from perspectives of OoD performance (Hendrycks et al., 2021), adversarial robustness (Goodfellow et al., 2014), and uncertainty estimation (Lakshminarayanan et al., 2016). See Appendix A for full details of the measurements used in this work.

# 4  EXPERIMENTAL RESULTS ON CIFAR

In this section, we report the results of sparse training with random pruning on CIFAR-10/100 under various evaluation measurements. For each measurement, the trade-off between sparsity and the corresponding metric is reported. Moreover, we also alter the depth and width of the model to check how the performance alters as the model size alters. The results are averaged over 3 runs. We report the results on CIFAR-10 of ResNet in the main body of this paper. The results on CIFAR-100 of

ResNet, and CIFAR-10/100 of VGG are shown in Appendix D and Appendix E, respectively. Unless otherwise stated, all the results are qualitatively similar.

## 4.1 PREDICTIVE ACCURACY OF RANDOM PRUNING ON CIFAR-10

We first demonstrate the performance of random pruning on the most common metric – test accuracy. To avoid overlapping of multiple curves, we share the results of GraSP in Appendix C. The main observations are as follows:

① **Performance of random pruning improves with the size of the network.** We vary the depth and width of ResNet and report the test accuracy in Figure 1. When operating on small networks, e.g., ResNet-20 and ResNet-32, we can hardly find matching subnetworks even at mild sparsities, i.e., 10%, 20%. With larger networks, e.g., ResNet-56 and ResNet-110, random pruning can match the dense performance at 60% $\sim$ 70% sparsity. Similar behavior can also be observed when we increase the width of ResNet-20 in Figure 2. See Appendix B for results on deeper models.

② **The performance difference between different pruning methods becomes indistinct as the model size increases.** Even though uniform sparsities fail to match the accuracy achieved by non-uniform sparsities (ERK and SNIP) with small models, their test accuracy raises to a comparable level as non-uniform sparsities with large models, e.g., ResNet-110 and ResNet-20-56.

③ **Random pruning with ERK even outperforms pruning with the well-versed methods (SNIP, GraSP).** Without using any information, e.g., gradient and magnitude, training a randomly pruned subnetwork with ERK topology leads to expressive accuracy, even better than the ones trained with the delicately designed sparsity ratios, i.e., SNIP, and GraSP (shown in Appendix C).

## 4.2 BROADER EVALUATION OF RANDOM PRUNING ON CIFAR-10

In general, sparse training with random pruning achieves quite strong results on CIFAR-10 in terms of uncertainty estimation, OoD robustness, and adversarial robustness without any fancy techniques. We summary main observations as below:

① **Randomly pruned networks enjoy better uncertainty estimation than their dense counterparts.** Figure 3 shows that randomly pruned ResNet-20 matches or even improves the uncertainty estimation of dense networks with a full range of sparsities. ECE of random pruning grows with the model size, in line with the finding of dense networks in Guo et al. (2017). Still, random pruning is capable of sampling matching subnetworks with high sparsities (e.g., 80%) except for the largest model, ResNet-110. Results of NLL are presented in Appendix G.2, where randomly pruned networks also match NLL of dense networks at extremely high sparsities.

② **Random pruning produces extremely sparse yet robust subnetworks on OoD.** Figure 4 plots the results of networks trained on CIFAR-10, tested on CIFAR-100. The results tested on SVHN are reported in Appendix G.1. Large network sizes significantly improve the OoD performance of random pruning. As the model size increases, random pruning can match the OoD performance of dense networks with only 20% parameters. Again, SNIP ratio and ERK outperform uniform sparsities in this setting.

③ **Random pruning improves the adversarial robustness of large models.** The adversarial robustness of large models (e.g., ResNet-56 and ResNet-110) improves significantly with mild sparsities in Appendix G.3. One explanation here is that, while achieving high clean accuracy, these over-parameterized large models are highly overfitted on CIFAR-10, and thus suffer from poor performance on adversarial examples (shown by Tsipras et al. (2019); Zhang et al. (2019) as well). Sparsity induced by random pruning serves as a cheap type of regularization which possibly mitigates this overfitting problem.

## 5 EXPERIMENTAL RESULTS ON IMAGENET

We have learned from Section 4 that larger networks result in stronger random subnetworks on CIFAR-10/100. We are also interested in how far we can go with random pruning on ImageNet (Deng et al., 2009), a non-saturated dataset on which deep neural networks are less over-parameterized than on CIFAR-10/100. In this section, we provide a large-scale experiment on ImageNet with various ResNets from ResNet-18 to ResNet-101 and Wide ResNet-50.

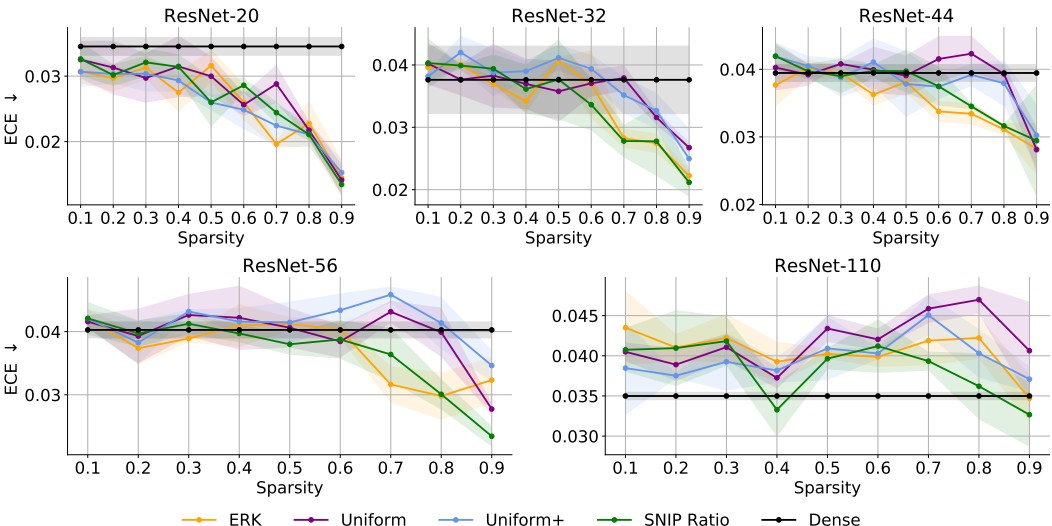

Figure 3: **Uncertainty estimation (ECE).** The experiments are conducted with various models on CIFAR-10. Lower ECE values represent better uncertainty estimation.

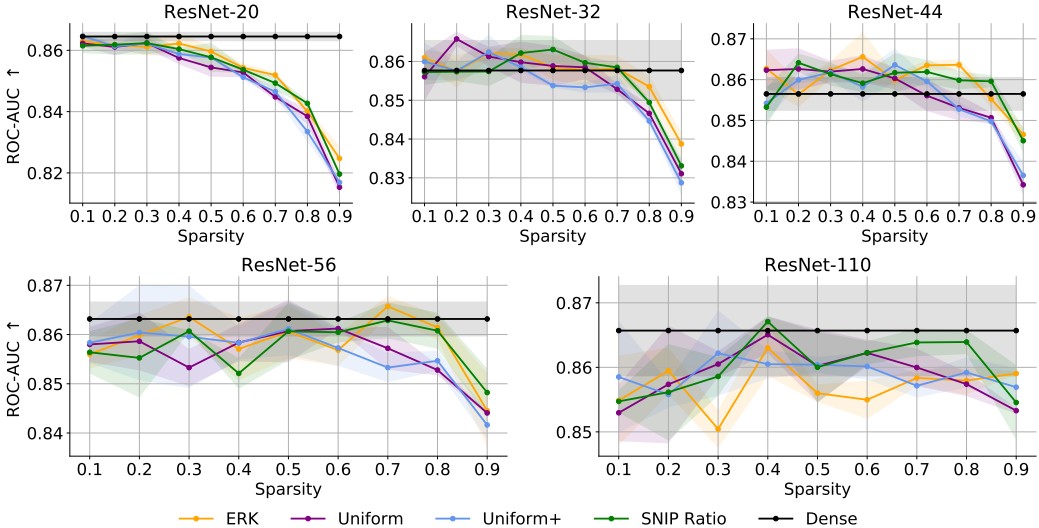

Figure 4: **Out-of-distribution performance (ROC-AUC).** Experiments are conducted with models trained on CIFAR-10, tested on CIFAR-100. Higher ROC-AUC refers to better OoD performance.

Same with CIFAR, we evaluate sparse training with random pruning from various perspectives, including the predictive accuracy, OoD detection, adversarial robustness, and uncertainty estimation. We choose SNIP and ERK+ sparsity ratios for Wide ResNet-50, and SNIP ratios for the rest of the architectures. The sparsity of randomly pruned subnetworks is set as [0.7, 0.5, 0.3]. We first show the trade-off between the parameter count and the test accuracy for all architectures in Figure 5-top-left. Moreover, to better understand the computational benefits brought by sparsity, we report the trade-off between test FLOPs and each measurement metric in the rest of Figure 5.

**Predictive Accuracy.** The parameter count-accuracy trade-off and the FLOPs-accuracy trade-off is reported in Figure 5-top-left and Figure 5-top-middle, respectively. Overall, we observe a very similar pattern as the results of CIFAR reported in Section 4. On smaller models like ResNet-18 and ResNet-34, random pruning can not find matching subnetworks. When the model size gets considerably larger (ResNet-101 and Wide ResNet-50), the test accuracy of random pruning quickly improves and matches the corresponding dense models (not only the small-dense models) at 30% ∼ 50% sparsities. While the performance difference between SNIP ratio and ERK on CIFAR-10/100 is

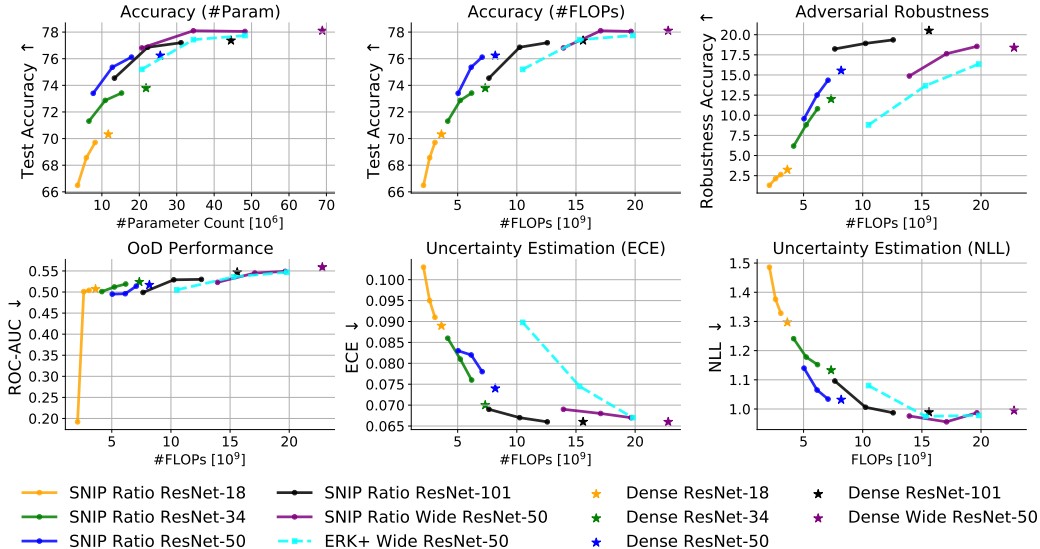

Figure 5: **Summary of evaluation on ImageNet.** Various Evaluation of ResNets on ImageNet, including predictive accuracy on the original ImageNet, adversarial robustness with FGSM, OoD performance on ImageNet-O, and uncertainty (ECE and NLL). The sparsity of randomly pruned subnetworks is set as [0.7, 0.5, 0.3] from left to right for each line.

vague, SNIP ratio consistently outperforms ERK+ with Wide Resnet-50 on ImageNet with the same number of parameters (see Appendix F for more details). Besides, we observe that random pruning receives increasingly larger efficiency gains (the difference between x-axis values of sparse models and dense models) with the increased model size on ImageNet.

While in Figure 5-top-left, a randomly pruned Wide ResNet-50 with SNIP ratios (purple line) can easily outperform the dense ResNet-50 (blue star) by 2% accuracy with the same number of parameters, the former requires twice as much FLOPs as the latter in Figure 5-top-middle. This observation highlights the importance of reporting the required FLOPs together with the sparsity when comparing two pruning methods.

**Broader Evaluation of Random Pruning on ImageNet.** As shown in the rest of Figure 5, the performance of the broader evaluation on ImageNet is extremely similar to the test accuracy. Random pruning can not discover matching subnetworks with small models. However, as the model size increases, it receives large performance gains in terms of other important evaluations, including uncertainty estimation, OoD detection performance, and adversarial robustness.

## 5.1 UNDERSTANDING RANDOM PRUNING VIA GRADIENT FLOW

The performance gap between the ERK and SNIP ratio on ImageNet raises the question – what benefits do the layer-wise sparsities of SNIP provide to sparse training? Since SNIP takes gradients into account when determines the layer-wise sparsities, we turn to gradient flow in the hope of finding some insights on this question. The form of gradient norm we choose is the effective gradient flow (Tessera et al., 2021) which only calculates the gradient norm of active weights. We measure the gradient norm of sparse networks generated by SNIP and ERK[2] during the early training phase. The results are depicted in Figure 6.

Overall, we see that the SNIP ratio indeed brings benefits to the gradient norm at the initial stage compared with ERK. Considering only SNIP (green lines) and ERK (orange lines), the ones with higher gradient norm at the beginning always achieve higher final accuracy on both CIFAR-10 and ImageNet. This result is on par with prior work (Wang et al., 2020), which claims that increasing the gradient norm at the beginning likely leads to higher accuracy. However, this observation does not

---

[2]The gradient norm patterns of ERK and ERK+ are too similar to distinguish. After specifically zooming in the initial training phase, we find that the gradient norm of ERK+ is slightly higher than ERK.

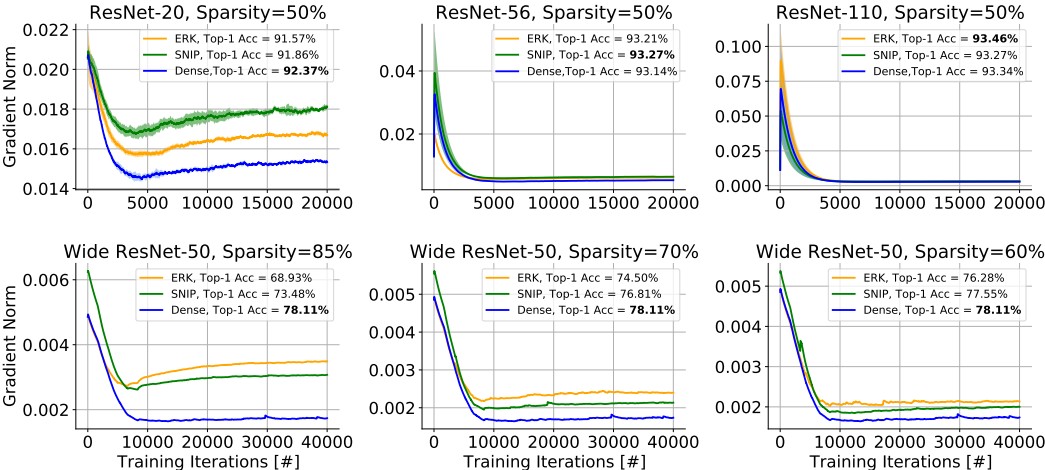

Figure 6: **Gradient norm of randomly pruned networks during training.** *Top:* Comparison between gradient norm of SNIP and ERK with 50% sparse ResNet-20, ResNet-56, and ResNet-110 on CIFAR-10. *Bottom:* Comparison between gradient norm of SNIP and ERK with Wide ResNet-50 on ImageNet at various sparsities.

hold for dense Wide ResNet-50 on ImageNet (blue lines). Even though dense models have a lower gradient norm than SNIP, they consistently outperform SNIP in accuracy by a large margin.

Instead of focusing on the initial phase, we note that the gradient norm early in the training (the flat phase after gradient norm drops) could better understand the behaviour of networks trained under different scenarios. We empirically find that final performance gaps between sparse networks and dense networks is highly correlated with gaps of gradient norm early in training. Training with small networks (e.g., ResNet-20) on CIFAR-10 results in large performance and gradient norm gaps between sparse networks and dense networks, whereas these two gaps simultaneously vanish when trained with large networks, e.g., ResNet-56 and ResNet-110. Similar behavior can be observed in Wide ResNet-50 on ImageNet in terms of sparsity. Both the performance and gradient norm gaps decrease gradually as the sparsity level drops from 85% to 60%. This actually makes sense, since large networks (either large model size or large number of parameters) are likely to still be over-parameterized after pruning, so that all pruning methods (including random pruning) can preserve gradient flow along with test accuracy equally well.

Our findings suggest that only considering gradient flow at initialization might not be sufficient for sparse training. More efforts should be invested to study the properties that sparse network training misses after initialization, especially the early stage of training (Liu et al., 2021a). Techniques that change the sparse pattern during training (Mocanu et al., 2018; Evci et al., 2020a) and weight rewinding (Frankle et al., 2020; Renda et al., 2020) could serve as good starting points.

## 6 CONCLUSION

In this work, we systematically revisit the underrated baseline of sparse training – random pruning. Our results highlight a counter-intuitive finding, that is, training a randomly pruned network from scratch without any delicate pruning criteria can be quite performant. With proper network sizes and layer-wise sparsity ratios, random pruning can match the performance of dense networks even at extreme sparsities. Impressively, a randomly pruned subnetwork of Wide ResNet-50 can be trained to outperforming a strong benchmark, dense Wide ResNet-50 on ImageNet. Moreover, training with random pruning intrinsically brings significant benefits to other desirable aspects, such as out-of-distribution detection, uncertainty estimation and adversarial robustness. Our paper indicates that in addition to appealing performance, large models also enjoy strong robustness to pruning. Even if we pruning with complete randomness, large models can preserve their performance well.

# 7 REPRODUCIBILITY

We have shared the architectures, datasets, and hyperparameters used in this paper in Section 3.2. Besides, the different measurements and metrics used in this paper are shared in Appendix A. We have released our code at https://github.com/VITA-Group/Random_Pruning.

# 8 ACKNOWLEDGEMENT

This work have been done when Shiwei Liu worked as an intern at JD Explore Academy. This work is partially supported by Science and Technology Innovation 2030 – "Brain Science and Brain-like Research" Major Project (No. 2021ZD0201402 and No. 2021ZD0201405). Z. Wang is in part supported by the NSF AI Institute for Foundations of Machine Learning (IFML).

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

## A    MEASUREMENTS AND METRICS

Formally, let's denote a (sparse) network $f : \mathcal{X} \to \mathcal{Y}$ trained on samples from distribution $\mathcal{D}$. $f(x, \theta)$ and $f(x, \theta \odot m)$ refers to the dense network and the pruned network, respectively. The test accuracy of the pruned subnetworks is typically reported as their accuracy on test queries drawn from $\mathcal{D}$, i.e., $\mathbb{P}_{(x,y) \sim \mathcal{D}}(f(x, \theta \odot m) = y)$. In addition to the test accuracy, we also evaluate random pruning from the perspective of adversarial robustness (Goodfellow et al., 2014), OoD performance (Hendrycks et al., 2021), and uncertainty estimation (Lakshminarayanan et al., 2016).

**Adversarial Robustness.**  Despite of the remarkable ability solving classification problems, the predictions of deep neural networks can often be fooled by small adversarial perturbations (Szegedy et al., 2013; Papernot et al., 2016). Prior works have shown the possibility of finding the sweet point of sparsity and adversarial robustness (Guo et al., 2018; Ye et al., 2019; Gui et al., 2019; Hu et al., 2020). As the arguably most naive method of inducing sparsity, we are also interested in if training a randomly pruned subnetwork can improve the adversarial robustness of deep networks. We follow the classical method proposed in Goodfellow et al. (2014) and generate adversarial examples with Fast Gradient Sign Method (FGSM). Specifically, input data is perturbed with $\epsilon \mathrm{sign}(\nabla_x \mathcal{L}(\theta, x, y))$, where $\epsilon$ refers to the perturbation strength, which is chosen as $\frac{8}{255}$ in our paper.

**Out-of-distribution performance.** The investigation of out-of-distribution (OoD) generalization is of importance for machine learning in both academic and industry fields. Since the i.i.d. assumption can hardly be satisfied, especially those high-risk scenarios such as healthcare, and military. We evaluate whether random pruning brings benefits to OoD. Following the classic routines (Augustin et al., 2020; Meinke & Hein, 2020), SVHN (Netzer et al., 2011), CIFAR-100, and CIFAR-10 with random Gaussian noise (Hein et al., 2019) are adopted for models trained on CIFAR-10; ImageNet-O as the OoD dataset for models trained on ImageNet.

**Uncertainty estimation.** In the security-critical scenarios, e.g., self-driving, the classifiers not only must be accurate but also should indicate when they are likely to be incorrect (Guo et al., 2017). To test effects of the induced sparsity on uncertainty estimation, we choose two widely-used metrics, expected calibration error (ECE) (Guo et al., 2017) and negative log likelihood (NLL) (Friedman et al., 2001).

## B    PREDICTIVE ACCURACY OF RESNET WITH VARYING WIDTH

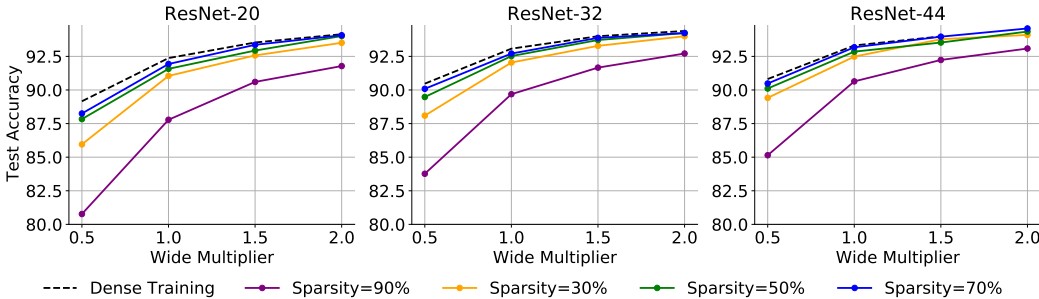

Figure 7: Test accuracy of training randomly pruned ResNet-20, ResNet-32, and ResNet-44 from scratch with ERK ratios when model width varies on CIFAR-10.

# C   PREDICTIVE ACCURACY OF RESNET ON CIFAR-10 INCLUDING GRASP

To avoid multiple lines overlap with each other, we report the results of GraSP ratio on CIFAR-10 separately in this Appendix. It is surprising to find that random pruning with GraSP ratio (red lines) consistently have lower accuracy than ERK and SNIP, except for extremely high sparsites, i.e., 0.8 and 0.9.

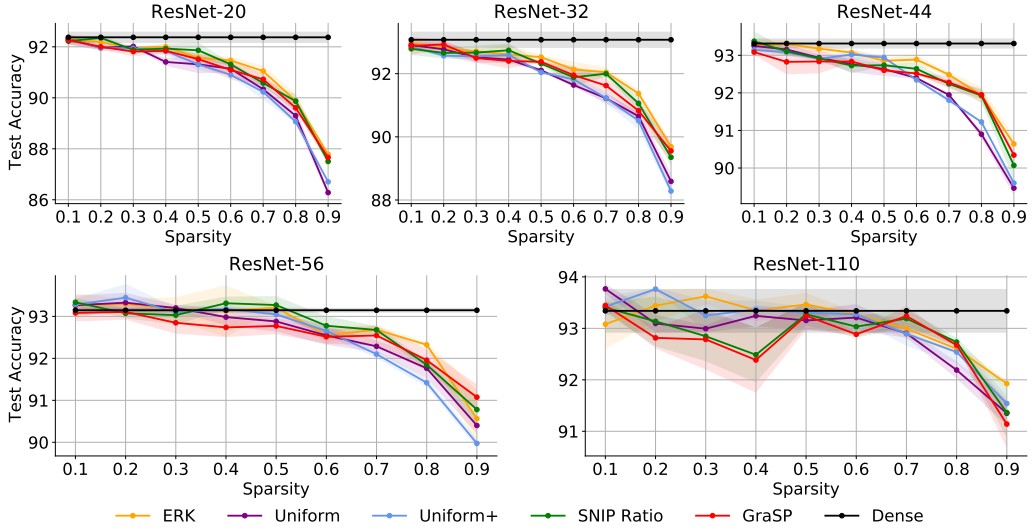

Figure 8: **From shallow to deep.** Test accuracy of training randomly pruned subnetworks from scratch with different depth on CIFAR-10. ResNet-A refers to a ResNet model with A layers in total.

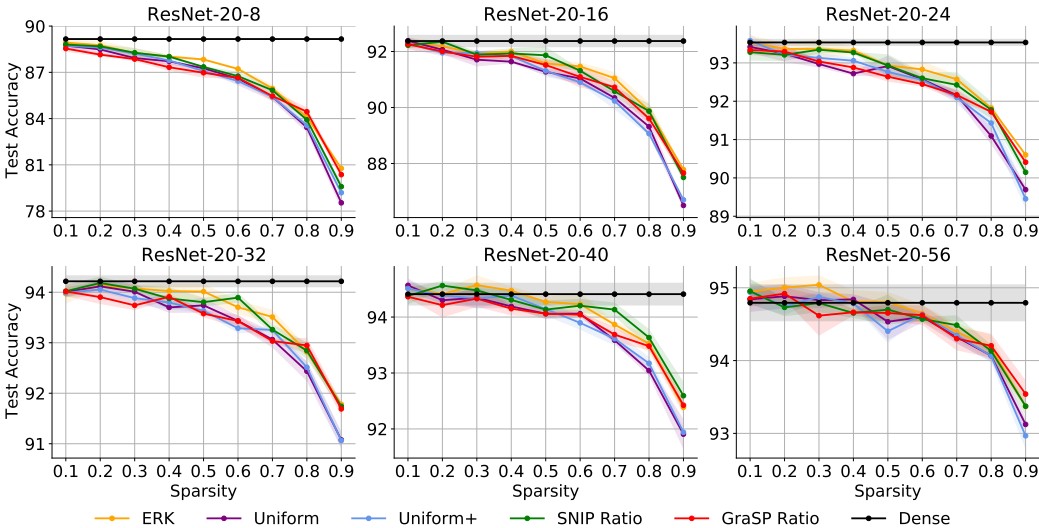

Figure 9: **From narrow to wide.** Test accuracy of training randomly pruned subnetworks from scratch with different width on CIFAR-10. ResNet-A-B refers to a ResNet model with A layers in total and B filters in the first convolutional layer.

## D  PREDICTIVE ACCURACY OF RESNET ON CIFAR-100

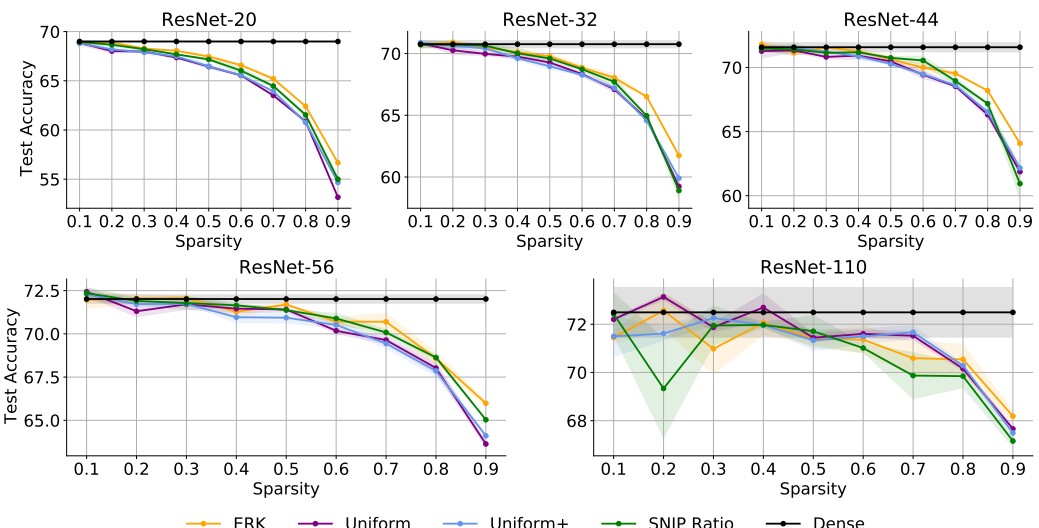

Figure 10: **From shallow to deep.** Test accuracy of training randomly pruned subnetworks from scratch with different depth on CIFAR-100. ResNet-A refers to a ResNet model with A layers in total.

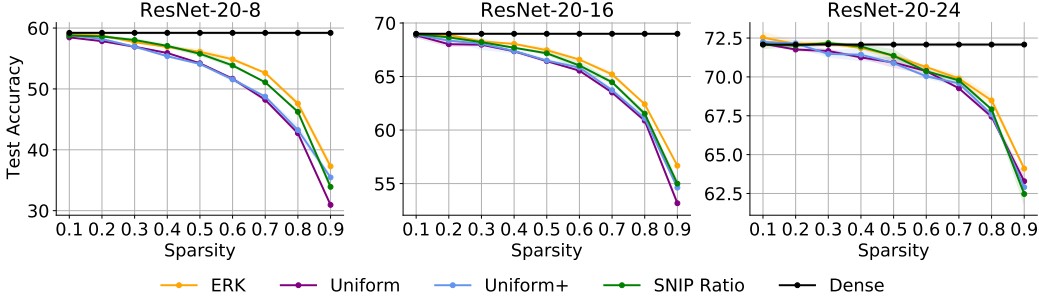

Figure 11: **From narrow to wide.** Test accuracy of training randomly pruned subnetworks from scratch with different width on CIFAR-100. ResNet-A-B refers to a ResNet model with A layers in total and B filters in the first convolutional layer.

# E PREDICTIVE ACCURACY OF VGG ON CIFAR-10/100

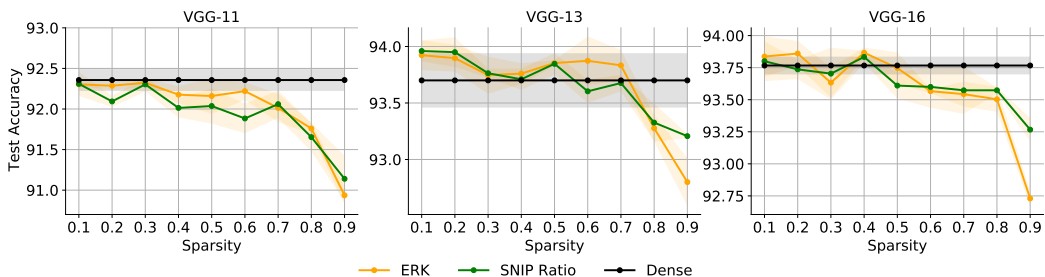

Figure 12: **From shallow to deep.** Test accuracy of training randomly pruned VGGs from scratch with different depth on CIFAR-10.

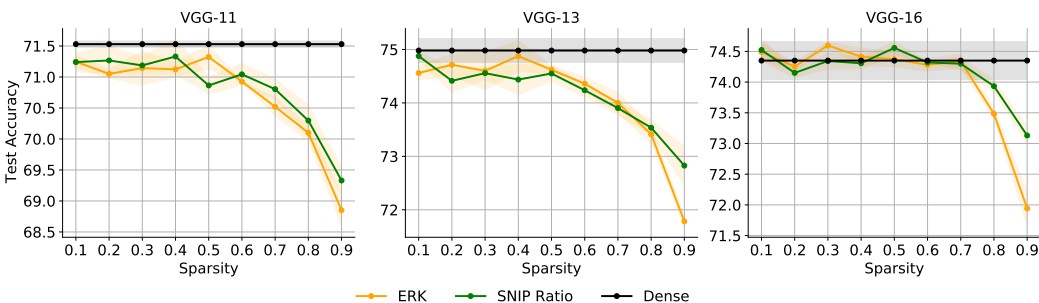

Figure 13: **From shallow to deep.** Test accuracy of training randomly pruned VGGs from scratch with different depth on CIFAR-100.

# F    COMPARING RANDOM PRUNING WITH ITS DENSE EQUIVALENTS WITH VARIOUS RATIOS.

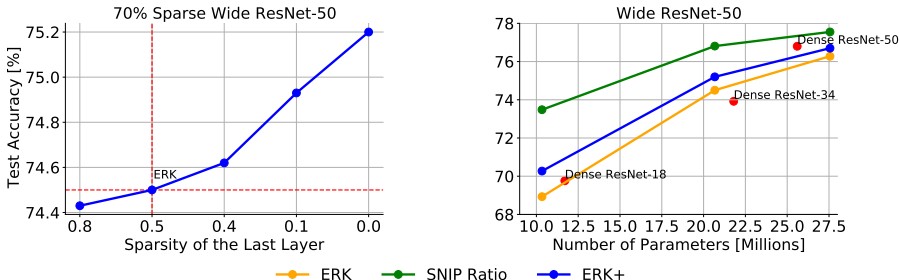

Figure 14: **Test accuracy of Wide ResNet-50 on ImageNet.** *Left:* Performance of ERK with various sparsity of the last fully-connected layer. We vary the sparsity of the last fully-connected layer while keeping the overall sparsity fixed as 70%. Results of the original ERK are indicated with dashed red lines. *Right:* Performance comparison between randomly pruned Wide ResNet-50 and the corresponding dense equivalents with a similar parameter count.

**ERK+.** We demonstrate the performance improvement caused by ERK+ on ImageNet. As mentioned earlier, ERK naturally allocates higher sparsities to the larger layers while allocating lower sparsities to the smaller ones. As a consequence, the last fully-connected layer is very likely to be dense for the datasets with only a few classes, e.g., CIFAR-10. However, for the datasets with a larger number of classes, e.g., ImageNet, the fully-connected layer is usually sparse. We empirically find that allocating more parameters to the last fully-connected layer while keeping the overall parameter count fixed leads to higher accuracy for ERK with Wide ResNet-50 on ImageNet.

We vary the last layer's sparsity of ERK while maintaining the overall sparsity fixed and report the test accuracy achieved by the corresponding sparse Wide ResNet-50 on ImageNet in Figure 14-left. We can observe that the test accuracy consistently increases as the last layer's sparsity decreases from 0.8 to 0. Consequently, we keep the last fully-connected layer of ERK dense for ImageNet and term this modified variant as ERK+.

**Compare random pruning with its dense equivalents.** To draw a more solid conclusion, we train large, randomly pruned Wide ResNet-50 on ImageNet and compare it to the dense equivalents with the same number of parameters on ImageNet. As shown in Figure 14-right, all randomly pruned networks outperform the dense ResNet-34 with the same number of parameters. ERK+ consistently achieves higher accuracy than ERK, even closely approaching the strong baseline – dense ResNet-50. More interestingly, the layer-wise sparsities discovered by SNIP boost the accuracy of sparse Wide ResNet-50 over dense ResNet-50, highlighting the importance of layer-wise sparsity ratios on sparse training . Given the fact that the performance gap between SNIP ratio and ERK on CIFAR-10 is somehow vague, our results highlight the necessity of evaluating any proposed pruning methods with large-scale models and datasets, e.g., ResNet-50 on ImageNet.

# G    BROADER EVALUATION OF RANDOM PRUNING ON CIFAR-10

## G.1    OUT-OF-DISTRIBUTION PERFORMANCE (ROC-AUC) .

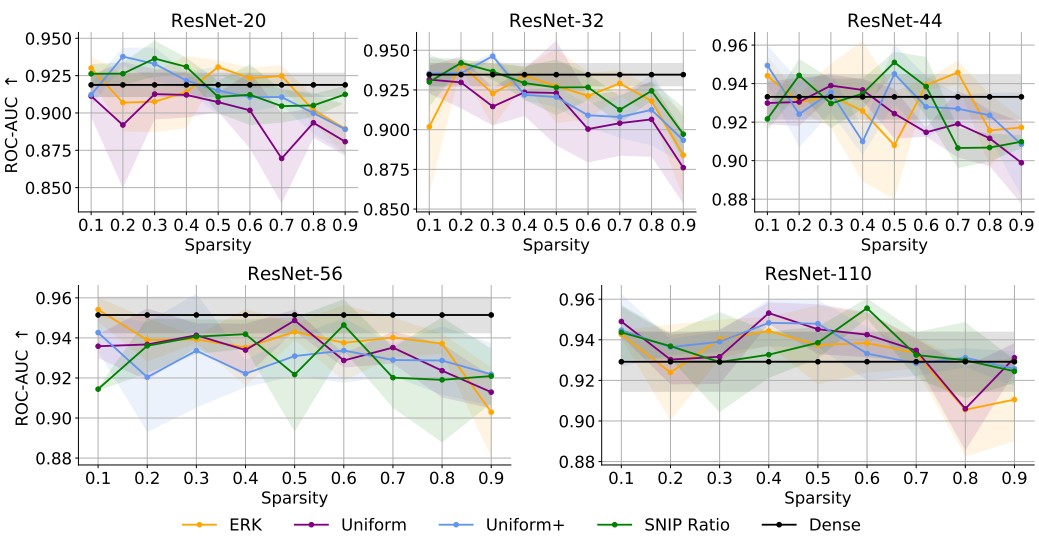

Figure 15: **Out-of-distribution performance (ROC-AUC).** The experiments are conducted with various models trained on CIFAR-10, tested on SVHN. Higher ROC-AUC refers to better OoD performance.

### G.2 UNCERTAINTY ESTIMATION (NLL)

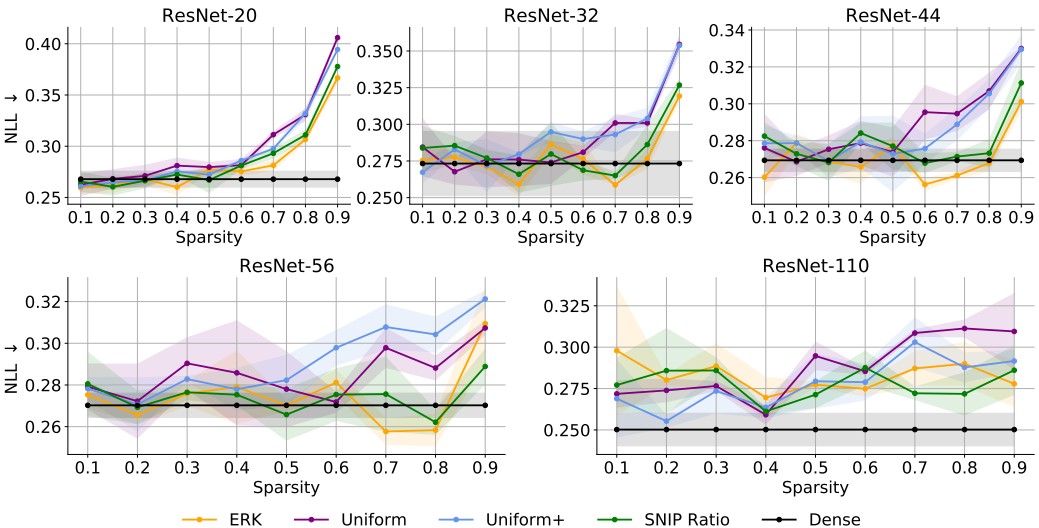

Figure 16: **Uncertainty estimation (NLL).** The experiments are conducted with various models on CIFAR-10. Lower NLL values represent better uncertainty estimation.

### G.3 ADVERSARIAL ROBUSTNESS

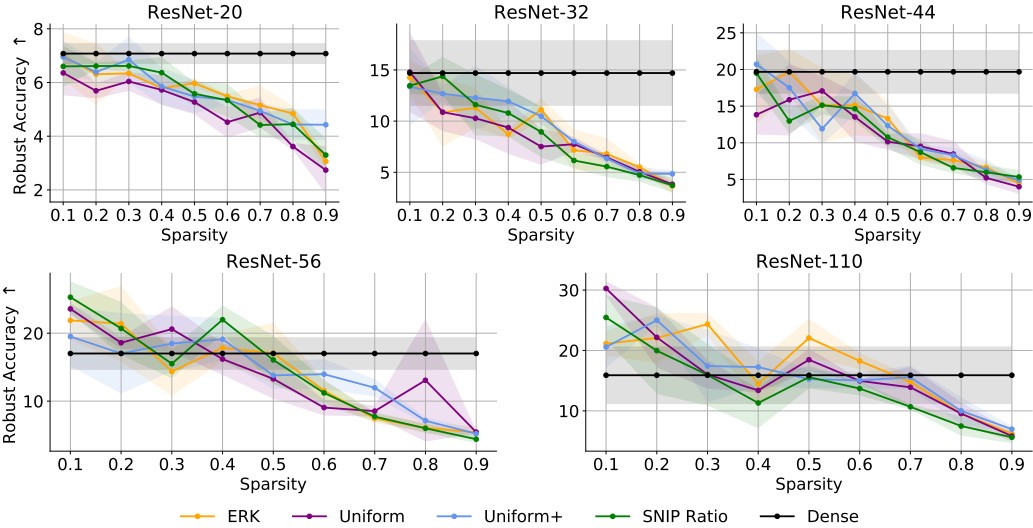

Figure 17: **Adversarial robustness.** The experiments are conducted with various models on CIFAR-10. Higher values represent better adversarial robustness.

