# OpenReview forum: "The Unreasonable Effectiveness of Random Pruning: Return of the Most Naive Baseline for Sparse Training"
_ICLR.cc/2022/Conference — ICLR 2022 Poster_

### Official Review · Reviewer_WXbx · 2021-10-16

**Correctness:** 4
**Technical Novelty And Significance:** 2
**Empirical Novelty And Significance:** 3
**Recommendation:** 6
**Confidence:** 5

**Main Review:**

I will go sequentially and start with strengths and proceed to weaknesses:

The writing of the is really good and I appreciate the authors for their efforts.
I also loved the extensive related work which will act as a great stepping stone for new researchers the same way Gale et al 2019 did in the past 2 years.

Strengths:
1) The revisiting of Random Pruning makes a lot of sense give the more complex pruning methods based on Hessians etc becoming a norm. The evaluation of this simple baseline should give perspective to new readers about what components affect the final deliverables most.
2) Very well crafted related work section
3) THe methodology is very clear and makes a clear distinction of where Random Pruning falls into the literature of learnable sparsity etc.,
4) The sparsity ratios being discussed are well explained are extensive.
5) The experimental setup is extremely clear and reproducible.
6) The results on CIFAR are extensive with various architectures and variants among them along with a very nice section 4.1 listing the observations that come of these evaluations.
7) While not as extensive similar observations were made on ImageNet with Wide ResNet50 as the base architecture.
8) The observations made on the importance of pruning ratios, the indifference of pruning methods at larger model sizes are valuable (while not novel).
9) The experiments trying to understand random pruning via gradient flow is interesting and give some perspective as to why certain pruning ratios might be doing better.
10) While only on CIFAR, I love the extent of broader evaluation on uncertainty, OOD, Robustness, etc for these random pruning models with different architectures and pruning ratios.
11) The plots are very clean and authors should be appreciated. The takeaways are explicitly mentioned making it easy for first-time readers.
12) The appendices are well laid out and detailed.

Weaknesses:
1) As I mentioned earlier, the method itself isn't novel and the certain observation of pruning larger models being better than dense counterparts of the same size isn't novel. Gale et al., 2019, Wallace et al 2020, Kusupati et al 2020 make the same observation but even Zhu & Gupta 2018 make similar observations and it is often observed that pruning large models is called large-sparse is better than small-dense. That is the same observation being made here.
2) The argument about the sparsity ratios being super important is not new either. There are papers like Su et al NeurIPS 2020 which argue that sparsity ratios are all that matter even in the LTH space and randomly changing the weights still gives them the tickets needed to win the lottery. This paper is missing in the related work but I am pretty sure (I am probably blanking out) there are papers that talk about the importance of sparsity ratios. Even papers like RigL and STR make a case for why sparsity ratios are important in general.
3) While on the topic of sparsity ratios, I also noticed that the benchmarked pruning results don't have inference cost involved in these cases. For example, I would love to see the FLOPs for Fig 4 (right) to see what the FLOPs are for pruned WRN vs a Dense ResNet.
4) I have no qualms about the broader evaluation except that I would love to see them on ImageNet even if it is for a single architecture. Is that something possible?

Overall, I think I don't see as much novelty in this paper (which is fine) however even the observations made are not completely novel. The evaluation is thorough and the observations are clean. However, I want to hear from other reviewers and the authors during the discussion before I move my decision towards acceptance. This is a great benchmarking paper, but at times looks like stitching across insights from other papers. Looking forward to the discussion.



**Summary Of The Paper:**

This paper revisits the OG and the simplest pruning baselines out there: Random Pruning. Random Pruning randomly zeros out the weights in a network with a target budget and is often used at a layer level based on the chosen sparsity ratios.

This paper is a detailed study into Random Pruning. As put forward in contributions, the discussion includes the dependence on network size, the importance of sparsity ratios, and lastly the general-purpose aspects like robustness and OOD, etc of these pruned networks.

The observations seem to point to the fact that as the model size increases the pruning method becomes obsolete and everything converges to the same solution. They also show that sparsity ratios matter more than anything else for pruning techniques. These two observations have been made in the past in different settings but still are valid observations.

Lastly, the paper extensively tries to evaluate the robustness aspects of these models + sparsity ratios for CIFAR.



**Summary Of The Review:**

As mentioned in the paper, the technique isn't novel however, the detailed study can benefit quite a few in the community given the potential ramifications of such observations. I would say it is a great benchmarking paper that would help people like "The State of Sparsity" (Gale et al., 2019) did, but again it doesn't make a super-strong case for acceptance. I am on the fence with a 5.  However, I am willing to change my mind after rebuttal and discussion with other reviewers.

I still think this paper is a good asset and will be a great one if the same scale of experiments is done on ImageNet instead of CIFAR and will prove to be a solid exploratory work in that case.

Overall, I am happy with the rebuttal. I want to see these updates and results in the main paper and not the appendix and you can do this as revision in the next few days.

I am increasing my score to 6 and will talk to other reviewers but push for an acceptance.

---

> ### Author Response · Authors · 2021-11-19
> **Response to the reviewer WXbx (2/2)**
>
> **Q3: I still think this paper is a good asset and will be a great one if the same scale of experiments is done on ImageNet instead of CIFAR and will prove to be a solid exploratory work in that case**
> - Thanks for your compliment and suggestions. We agree that the same scale of experiments on ImageNet is important. We have expanded the experiments on ImageNet with various architectures from ResNet-18 to ResNet-101 and Wide ResNet-50. Due to the limited time and resources, we choose the SNIP and ERK+sparsity ratios for Wide ResNet-50 and only SNIP ratios for the rest of the architectures. The sparsity of randomly pruned subnetworks is set as [0.7, 0.5, 0.3] from left to right.
>
> - The results are shared in Appendix E. Overall, we observe a very similar pattern as illustrated in the original submission. On smaller models like ResNet-18 and ResNet-34, random pruning can not find matching subnetworks. When the model size gets considerably larger (ResNet-101 and Wide ResNet-50), random pruning can match the dense models at 50% sparsity.
>
>
> **Q4: While on the topic of sparsity ratios, I also noticed that the benchmarked pruning results don't have inference cost involved in these cases. For example, I would love to see the FLOPs for Fig 4 (right) to see what the FLOPs are for pruned WRN vs a Dense ResNet.**
> - We provide a scaled-up overview of the trade-off between accuracy and test FLOPs with various models in Appendix F. We can see that the number of FLOPs required by randomly pruned Wide ResNet-50 is still larger than the one required by the dense ResNet-50. Yet, the corresponding performance gain is also notable (2%). More importantly, when the model size is large enough, the accuracy of the randomly pruned sparse networks quickly grows and matches the corresponding large dense network (not only the small-dense models).
>
> **Q5: I have no qualms about the broader evaluation except that I would love to see them on ImageNet even if it is for a single architecture. Is that something possible?**
> - As you suggested, we added the same evaluation on ImageNet as well in Appendix G. The performance on the broader evaluation on ImageNet is extremely similar to the test accuracy.  Random pruning can not discover matching subnetworks with small models. However, as the model size increases, it receives large gains in terms of other important evaluations, including uncertainty estimation, OoD robustness, and adversarial robustness.

---

> > ### Comment · Reviewer_WXbx · 2021-11-19
> > **Thanks for the clarifications (Part 2)**
> >
> > 3) I like the new results a bunch. The text and captions for the figures can be much more information. I would like to see all these results in the main paper and with all the observations and explanations being presented here.
> >
> > 4) Again, same as the above thing. I love the results and the fact you acknowledge what is lacking. Please push them to the main paper.
> >
> > 5) Thanks for these additional results, these help me ensure that the results are correct across all the board.
> >
> > Overall, I am happy with the rebuttal. I want to see these updates and results in the main paper and not the appendix and you can do this as revision in the next few days.
> >
> > I am increasing my score to 6 and will talk to other reviewers but push for an acceptance.

---

> > > ### Author Response · Authors · 2021-11-21
> > > **New revision including all the results in the main paper**
> > >
> > > We really appreciate your acceptance. Your support means a lot to us!
> > >
> > > We re-arranged the paper as you advised to fit all the results in the main paper while being well-organized and smooth.
> > >
> > > Please let us know if you have additional advises or changes.

---

> ### Author Response · Authors · 2021-11-19
> **Response to the reviewer WXbx (1/2)**
>
> We sincerely appreciate your positive comments overall and the very detailed summary of our strengths. We are glad that you find our paper is as useful as the insightful and solid work “The State of Sparsity” (Gale et al., 2019). We have provided a general response to address your novelty concern above. And below, you can find our detailed response to your comments in-line below.
>
> **Q1: As I mentioned earlier, the method itself isn't novel and the certain observation of pruning larger models being better than dense counterparts of the same size isn't novel. Gale et al., 2019, Wallace et al 2020, Kusupati et al 2020 make the same observation but even Zhu & Gupta 2018 make similar observations and it is often observed that pruning large models is called large-sparse is better than small-dense. That is the same observation being made here.**
> - We fully agree that large-sparse performs better than small-dense has been shown in many prior works. However, they mainly focus on pruning after training (Wallace et al 2020) and during training (Gale et al., 2019) with non-random pruning. It remains unknown if we can still find matching subnetworks only by random pruning at initialization.
> Moreover, our finding is not limited to small-dense but the large-dense with the same architecture. Due to its ultra training efficiency, proving the feasibility of this setting is very promising. However, the performance of pruning at initialization is relatively unsatisfactory. According to the results of Figure 4 in [Frankle et al. 2020b](https://openreview.net/forum?id=Ig-VyQc-MLK), pruning ResNet-50 at the initialization with all sparsities even the trivial sparsity, i.e., 10%, 20%, can not match the accuracy achieved by the dense ResNet-50. This finding will significantly degrade the value of pruning at initialization or sparse training.
> In this paper,  we show for the first time that random pruning performs very differently on large modes (e.g., Resnet-101, Wide ResNet-50) and present compelling results that have been never explored before, i.e.,  training a 50% randomly pruned Wide ResNet-50 is sufficient to match the accuracy of dense Wide ResNet-50.
>
> **Q2: The argument about the sparsity ratios being super important is not new either. There are papers like Su et al NeurIPS 2020 which argue that sparsity ratios are all that matter even in the LTH space and randomly changing the weights still gives them the tickets needed to win the lottery. This paper is missing in the related work but I am pretty sure (I am probably blanking out) there are papers that talk about the importance of sparsity ratios. Even papers like RigL and STR make a case for why sparsity ratios are important in general.**
>
> - We can not agree more that the sparsity ratio is crucial is not novel. We can observe this phenomenon in many works. We didn’t claim that our paper is the first one to identify that the sparsity ratio is an important aspect in randomly connected static sparse neural networks, but our phrasing may be misleading. We apologize for that.
> - The main goal of our paper is to comprehensively study the performance of random pruning at initialization. While prior works (Su et al, 2020; Frankle et al. 2020b) have observed that random pruning can be more competitive in certain cases, they did not give principled guidelines on when and how it can become that good; nor do they show it can match the performance of dense networks on ImageNet.  Standing on the shoulders of giants, our work summarized principles by more rigorous studies, and demonstrate the strongest result so far, that randomly pruned sparse Wide ResNet-50 can be sparsely trained to outperform a dense Wide ResNet-50, on ImageNet.
> - Moreover, compared with the ad-hoc sparsity ratios used in Su et al, 2020, we show that ERK (Evci et al 2020a) and our modified ERK+ are more general sparsity ratios that consistently demonstrate competitive performance without careful layer-wise sparsity design for every architecture. Specifically, ERK+ ratio achieves similar accuracy with the dense Wide ResNet-50 on ImageNet while being data free, feedforward free, and dense initialization free. We have added a paragraph of careful discussion with Su2020 to clarify our similarities and dissimilarities in the revision.

---

> > ### Comment · Reviewer_WXbx · 2021-11-19
> > **Thanks for the clarifications**
> >
> > Dear authors,
> >
> > Thanks for the detailed rebuttal and paper updates. The clarifications on the contributions are clearer and the experiments and insights are much more valuable with the scale of the experiments. I am not sure what stopped you from doing it for the actual submission itself.
> >
> > Anyway, let me try to respond to the rebuttal sequentially.
> >
> > 1) The clarification helps, hope the writing is made clearer in the next version. Also, you can change the main text in the revisions and I would love to see the updated manuscript not just added appendices.
> >
> > 2) Thanks for the clarification on the contributions again. This helps, but I would again like to see updated working in the main paper before the final camera-ready.
> >
> > Continued on next comment.

---

### Official Review · Reviewer_nAhU · 2021-11-01

**Correctness:** 4
**Technical Novelty And Significance:** 3
**Empirical Novelty And Significance:** 4
**Recommendation:** 8
**Confidence:** 4

**Main Review:**

Strengths:
-	This paper systematically revisits the underrated baseline of sparse training – random pruning. The authors present a series of very interesting findings with solid experimental supports.
-	The findings first highlight a hidden gem role of larger models – their high tolerance to sparse training. As the networks grow wider and deeper, the performance of training a randomly pruned sparse network will quickly grow to matching that of its dense equivalent, even at high sparsity ratios. This is an inspiring new insight of broad potential impact in current ML trend, especially how to train larger models more efficiently.
-	Another meaningful discovery is that, echoing prior arts, the layer-wise sparsity ratio or “effective width” seems to be a critical factor for high performant sparse training. Borrowing ratios from SNIP could lead to pushing the performance of a completely random sparse Wide ResNet-50 over the strong baseline of densely trained ResNet-50 on ImageNet. However, this part lacks deep understanding of its underlying cause – see first bullet under “questions”.
-	The effectiveness of ERK on ImageNet (with the help of sparsity ratios) is surprising and impressive – perhaps the strongest I’ve seen in relevant empirical literature. It can outperform SNIP/Grasp for many cases, both in accuracy and in robustness/other aspects.
-	Codes are included as supplementary, and experiments have included sufficient details to be reproducible.


Questions or Suggestions:
-	One weakness I feel about this paper is its interpretation part in Section 4.3. Especially, I find it hard to comprehend what “early-stage gradient gaps” mean or can reflect for sparse network optimization. The authors basically described the phenomenon then stopping without digging more insights.
-	Moreover, if early training dynamics are indeed essential for improving sparse training, can this be combined with dynamic sparse training (e.g. Evci 2020a)? As one possible way to balance randomness with learned connectivity, could there be some dynamic sparsity ratio re-allocation during training? The authors are encouraged to discuss more.
-	A minor limitation of this work is that random sparsity will only work so well when the network is large/wide enough, as the authors also pointed out.
-	(Optional) With SNIP/Grasp being evaluated, it is recommended that the authors can compare with SynFlow and some more recent sparse training baselines too. Including comparison to RigL would be interesting too.


**Summary Of The Paper:**

Sparse training has traditionally relied on carefully crafted sparse patterns or pruning criteria. This paper reports a novel discovery that without any delicate pruning criteria or careful sparsity structures, training a randomly pruned sparse network from scratch can match the performance of its dense equivalent.

**Summary Of The Review:**

The work questions the common wisdom of sparse training and found properly trained randomly pruned sparse networks from scratch can match dense networks in many ways. This finding is counter-intuitive yet important. The paper could yet be strengthened further by providing more interpretations of its observations.

---

> ### Author Response · Authors · 2021-11-19
> **Response to the reviewer nAhU**
>
> We are grateful for your support and detailed comments! We provide responses to address your comments below.
>
> **Q1: One weakness I feel about this paper is its interpretation part in Section 4.3. Especially, I find it hard to comprehend what “early-stage gradient gaps” mean or can reflect for sparse network optimization. The authors basically described the phenomenon then stopping without digging more insights; Borrowing ratios from SNIP could lead to pushing the performance of a completely random sparse Wide ResNet-50 over the strong baseline of densely trained ResNet-50 on ImageNet. However, this part lacks a deep understanding of its underlying cause.**
>
> - Thanks for your questions. The “gradient norm early in the training” refers to the training phases where the decrease of gradient norm stops in Figure 5 (e.g., after around 10000 iterations with Wide ResNet-50 on ImageNet). We try to understand the success of SNIP on ImageNet from the perspective of gradient norm since SNIP takes gradient into consideration compared with ERK. Prior work on gradient flow [1] claims that a higher gradient norm at the initialization is more appealing for sparse training. We find this claim can not generalize to the dense network training. Instead, our results indicate that only considering gradient flow at initialization might not be sufficient for sparse training. The gradient norm gap between sparse networks and dense networks at the early phase could be more important. The correlation between the gradient norm gap and behaviors of different methods is clear and consistent (agreed by Reviewers GL4s and WXbx). It helps explain why ERK and SNIP ratios perform equally well on CIFAR-10 whereas SNIP ratio significantly outperforms ERK with Wide ResNet-50 on ImageNet. Moreover, it makes good connections with previous works that analyze the early stage of training.
>
>
> **Q2: Moreover, if early training dynamics are indeed essential for improving sparse training, can this be combined with dynamic sparse training (e.g. Evci 2020a)? As one possible way to balance randomness with learned connectivity, could there be some dynamic sparsity ratio re-allocation during training? The authors are encouraged to discuss more.**
>
> - Yes, the early training phase is indeed essential to (dynamic) sparse training. One effective way to balance randomness with learned connectivity at the beginning phase is dynamic sparsity training [2,3]. By pruning unimportant connections and adding new connections back, dynamic sparse training can jointly optimize the sparse connectivity and model weights. It has been shown in [4] that DST can further optimize the learned sparse connectivity by SNIP and LT, improving the performance significantly.
>
> - Even for DST, the early training phase is crucial. [5] recently shows that starting from a relatively denser subnetwork (sparsity=50%) rather than a fixed target sparsity, RigL (Evci 2020a) achieves a large performance improvement while discovering more efficient sparsity ratios than ERK. The larger (denser) initial subnetwork provides a larger exploration space for dynamic sparse training at the early training phase.
>
>
>
> [1] [Wang, Chaoqi, et al. "Picking winning tickets before training by preserving gradient flow." arXiv preprint arXiv:2002.07376. ICLR 2020.](https://arxiv.org/abs/2002.07376)
>
> [2] [Mocanu, Decebal Constantin, et al. "Scalable training of artificial neural networks with adaptive sparse connectivity inspired by network science." Nature communications 9.1 (2018): 1-12.](https://www.nature.com/articles/s41467-018-04316-3)
>
> [3] [Evci, Utku, et al. "Rigging the lottery: Making all tickets winners." International Conference on Machine Learning. PMLR, 2020.](https://arxiv.org/abs/1911.11134)
>
> [4] [Liu, Shiwei, et al. "Do we actually need dense over-parameterization? in-time over-parameterization in sparse training." arXiv preprint arXiv:2102.02887. ICML 2021.](https://arxiv.org/abs/2102.02887)
>
> [5] [Liu, Shiwei, et al. "Sparse Training via Boosting Pruning Plasticity with Neuroregeneration." arXiv preprint arXiv:2106.10404. NeurIPs 2021.](https://arxiv.org/abs/2106.10404)

---

### Official Review · Reviewer_Gr4w · 2021-11-02

**Correctness:** 3
**Technical Novelty And Significance:** 2
**Empirical Novelty And Significance:** 2
**Recommendation:** 6
**Confidence:** 5

**Main Review:**

Strengths:

1.	As said in the paper, random pruning is the very baseline for all pruning methods. Its performance was believed to be quite bad in the past, while this paper shows evidence on the contrary. This appears quite surprising and interesting.
2.	They have extensive experiments trying to support their claim.
3.	They further show random pruning networks can outperform dense counterparts in other favorable aspects. Potentially, this can make random pruning more useful in practice.

Weaknesses:

This paper appears to present surprising results (as indicated by the title -- The “Unreasonable” Effectiveness). However, the claims and contributions in this paper need to be rectified and re-evaluated.

First, a more rigorous way to treat the terminologies is needed. Pruning can be split into pruning a pretrained model (which is the traditional way), and pruning at initialization (i.e, pruning a randomly initialized network). What this paper really discussed and compared is pruning at initialization (PaI). PaI is only a relatively small track in pruning currently. And importantly, it has been shown PaI seriously underperforms the traditional pruning (pruning a pretraiend model), see [*1]. So the title should be “The Unreasonable Effectiveness of Random Pruning at Initialization”, not the most general pruning case.

Otherwise, if you want to claim random pruning can be on par with other pruning methods (let’s take the most simple magnitude pruning as example) under the traditional pruning case, please try to add this result: Prune a pretrained ResNet50 with 90% sparsity on ImageNet, compare random pruning with magnitude pruning (extensive previous methods actually have shown the latter is better).
Then, if we are talking about pruning at initialization, the so-called “unreasonable effectiveness” is actually not that unreasonable, provided you’ve seen [*1], which is cited as Frankle et al. (2020b) in the paper but I don’t think it is well discussed. [*1] has quite a lot of overlap with this paper.

Specifically, in [*1], they propose several sanity-check rules to evaluate whether a kind of pruning criterion is really effective or not: Randomly shuffling, Reinitialization, Inversion, in their Sec. 5. Particularly, for the random shuffling, they shuffled the masks obtained by some PaI methods (like SNIP, SynFlow, GraSP) and found  after mask random shuffling, these methods can achieve comparable accuracy (“All methods maintain accuracy or improve when randomly shuffled”). Randomly shuffling masks is essentially the same as random pruning. That is, in [*1], it has been shown that random pruning can perform on par with other PaI methods like SNIP. So the major point of this paper is actually reiterating what has been found in [*1].

Even similarly, in [*1], after discovering randomly shuffling masks does not hurt accuracy, they thus proposed “In other words, the useful information these techniques extract is not which individual weights to remove, but rather the layerwise proportions by which to prune the network”. Then look at the 2nd contribution claimed in this paper: “We further identify that appropriate layer-wise sparsity ratios can be an important booster for the performance of random pruning”. They are basically the same.
Given above, this paper has non-trivial overlaps with [*1]. Although it presents more empirical evidence to show random pruning can be useful in some cases (out-of-distribution detection, uncertainty estimation, and adversarial robustness), yet the major claims and points are already established in [*1].

Also, in the experiments, some are questionable. E.g., they compare pruned WideResNet50 to ResNet50 (Sec. 4.2). Although they have similar #params, notably they are two different networks. The comparison is not an apple-to-apple comparison (not scientifically valid), no matter how surprising the result may look. To claim random pruning is effective, what should be done is to compare randomly pruned WideResNet50 to a dense WideResNet50, not ResNet50. If the authors can provide this result and still show they have similar accuracy, I will improve my rating.

[*1] Pruning Neural Networks at Initialization: Why are We Missing the Mark?. ICLR, 2021.


**Summary Of The Paper:**

This paper discusses pruning at initialization (PaI) and they argue random pruning can be quite strong actually, whose performance was under-rated in the past. They present abundant experiments to show random pruning can perform on par with other PaI methods with dedicated pruning criteria. They further show random pruning networks can outperform dense counterparts in other favorable aspects, such as out-of-distribution detection, uncertainty estimation, and adversarial robustness.

**Summary Of The Review:**

This paper has substantial overlap with a previous paper. Also, they have unfair comparisons in their experiments.

---

> ### Author Response · Authors · 2021-11-19
> **Response to the reviewer Gr4w (2/2)**
>
> **Q2: Frankle et al. 2020b has quite a lot of overlap with the paper.**
> - Thanks for pointing out this important and insightful prior work which we are actually quite familiar with it. While there are indeed some overlaps existing in these two papers, we argue with confidence that our main paper differs from Frankle et al. 2020b from motivation, the main study, and observations.
>
> - **Motivation**: The main purpose of  Frankle et al. 2020b [*1], as presented in the paper, is to “assess the efficacy of these pruning methods at initialization”. They use several sanity-check rules (including random pruning) to show that the current PaI methods actually prune layers, not specific weights, whereas the magnitude pruning does not share this behavior, reflecting a broader challenge inherent
> to pruning at initialization. Thus, according to [*1]  random pruning at initialization is the weakest method while pruning after training is the strongest one.
> On the contrary, we highlight to the reviewer that our goal is exactly the opposite of [*1], i.e., studying the effectiveness of random pruning. While prior works (Su et al, 2020; Frankle et al. 2020b) have observed that random pruning can be more competitive in certain cases, they did not give principled guidelines on when and how it can become that good; nor do they show it can match the performance of dense networks on ImageNet. Standing on the shoulders of giants, our work summarized principles by more rigorous studies, and demonstrate the strongest result so far, that randomly pruned sparse Wide ResNet-50 can be sparsely trained to outperform a dense Wide ResNet-50, on ImageNet.
>
>
> - **The main study**:  To fully understand the necessary conditions for the success of random pruning. We comprehensively study the effect of various components, including, sparsity ratios, model sizes, and other practical evaluations. The last two research aspects are out of the scope of Frankle et al. 2020b.
>
> - **Observations**:  More importantly, our paper shows very different findings from Frankle et al. 2020b.
> We show that the ERK ratio and our modified version ERK+ ratio are very competitive with the well-versed SNIP and GraSP across various models, while being data-independent and requiring no extra information such as gradient and hessian. Specifically, we show that ERK+ achieves similar performance with the dense Wide ResNet-50 on ImageNet. Compared with the ad-hoc ratios used in Su et al 2020, ERK+ does not require ad-hoc ratios specifically designed for each different network.
>
> - Even though Frankle et al. 2020b show that randomly shuffled subnetworks can maintain or improve the original PaI methods, the accuracy achieved by random pruning on ImageNet is quite unsatisfactory. According to the results of Figure 4 in Frankle et al. 2020b, randomly pruned ResNet-50 at all sparsities even the trivial sparsity, i.e., 10%, 20%, can not match the accuracy achieved by the dense ResNet-50. This finding will significantly degrade the value of pruning at initialization or sparse training. On the contrary, our findings show that when the model size is large enough, training a 50% randomly pruned Wide ResNet-50 is sufficient to match the dense Wide ResNet-50.  We believe this finding is very important for the ML community.
>
>
> **Q3: Also, in the experiments, some are questionable. E.g., they compare pruned WideResNet50 to ResNet50 (Sec. 4.2). Although they have similar #params, notably they are two different networks. The comparison is not an apple-to-apple comparison (not scientifically valid), no matter how surprising the result may look. To claim random pruning is effective, what should be done is to compare randomly pruned WideResNet50 to a dense WideResNet50, not ResNet50. If the authors can provide this result and still show they have similar accuracy, I will improve my rating.**
>
> - We first explain that this comparison follows the convention of many prior works, e.g. [edgepopup] (https://openaccess.thecvf.com/content_CVPR_2020/papers/Ramanujan_Whats_Hidden_in_a_Randomly_Weighted_Neural_Network_CVPR_2020_paper.pdf) (who compared sparse wide ResNet-50 with dense ResNet-34), among others. In this vein, we feel our way to compare (sparse Wide ResNet-50 > dense ResNet-50) will clearly show our “surprising” advances in a context.
> - Second, we follow your advice to present the results of sparse versus dense Wide ResNet-50 in the following table. Indeed, we can even achieve comparable accuracies to dense Wide ResNet-50 under a range of sparsities. Consequently, we humbly ask you to reconsider improving the rating of our paper, as you generously promised.
> |  Methods  | | Wide ResNet-50  |ImageNet |  |
> | ------------- | :-----------: |:-----------:|:-------------:|:-------------:|
> | Sparsity | 0.85 | 0.70 | 0.60 | 0.50 |
> | ERK+ | 70.27 | 75.20 | 76.70 | 77.43|
> | SNIP | 73.48 | 76.81 | 77.55 | 78.10 |
> | Dense | 78.11 | 78.11 | 78.11 | 78.11 |

---

> > ### Comment · Reviewer_Gr4w · 2021-11-27
> > **After rebuttal**
> >
> > Thanks for the responses!
> >
> > The authors said because a previous paper compared sparse WideResNet50 to ResNet34, so they do the same thing. Honestly, I do not think this argument is strong still. How a previous paper conducted the comparison does not really lend any advantage here. Unfair comparison is unfair per se. Especially, the pruned WRN-50 has around 2x FLOPs than ResNet50 (see Fig. 5, upper middle). It is not clear that, if I enlarge ResNet50 to the same FLOPs as the pruned WRN-50 (I believe #FLOPs is of more interest than #Params nowadays), how would they compare against each other? If possible, I strongly suggest the authors add this experiment in the future.
> >
> > But in general, the authors convinced me that this work is different from Frankle et al. (2020b) and shows empirical improvement than the random pruning in Frankle et al. (2020b). I thus increased my rating to 6.

---

> > > ### Author Response · Authors · 2021-11-28
> > > **Thanks for your timely reply and acceptance!**
> > >
> > > Dear Reviewer Gr4w,
> > >
> > > We fully agree that comparison between dense ResNet-50 with the pruned WRN-50 is unfair, especially given the large FLOPs difference. And we highly appreciate that you gave us the opportunity to support our claim by adding the comparison between randomly pruned WRN-50 with the dense WRN-50. With this fair comparison, we show that randomly pruned WRN-50 can match the performance of dense WRN-50, supporting the effectiveness of random pruning at initialization.
> > >
> > > We sincerely thank you for your review and valuable comments again!
> > >
> > > Authors

---

> ### Author Response · Authors · 2021-11-19
> **Response to the reviewer Gr4w (1/2)**
>
> Many thanks for reviewing our paper. We provide pointwise responses to your concerns.
>
> **Q1: “First, a more rigorous way to treat the terminologies is needed. Pruning can be split into pruning a pre-trained model (which is the traditional way), and pruning at initialization (i.e, pruning a randomly initialized network). What this paper really discussed and compared is pruning at initialization (PaI). PaI is only a relatively small track in pruning currently. And importantly, it has been shown PaI seriously underperforms the traditional pruning (pruning a pretraiend model). So the title should be “The Unreasonable Effectiveness of Random Pruning at Initialization”, not the most general pruning case.”**
>
> - We never claim this work to compare with traditional pruning, but rather sparse training is the focus. We focus on the most efficient sparse-to-sparse training, and random pruning is the naive baseline to obtain the initial sparsity to kick off that. Therefore, we prefer to keep the current title as we believe it faithfully reflects our theme. But we agree with your suggestion and rephrase the abstract to differentiate our work more clearly from traditional pruning. For example, we rewrite the first sentence in the abstract as "Random pruning is arguably the most naive way to attain sparsity in neural networks, but has been deemed uncompetitive by either post-training pruning or sparse training. In this paper, we focus on sparse training and highlight a perhaps counter-intuitive finding, that random pruning at initialization can be quite powerful for the sparse training of modern neural networks". We also make other revisions accordingly in the introduction, such as “While random pruning is a universal method that can happen at any stage of training, training a randomly pruned network from scratch is arguably the most appealing way, owing to its “end-to-end” saving potential for the entire training process besides the inference. Due to this reason, we focus on random pruning at initialization (or sparse training) in this paper.”, etc. We hope those revisions have clarified your confusion and concerns.

---

> ### Author Response · Authors · 2021-11-24
> **Response to the reviewer Gr4w**
>
> Dear Reviewer Gr4w,
>
> We thank for your review time, and we really hope to have a further discussion with you to see if our response solves the concerns.
>
> For example, as we showed in the response, randomly pruned WideResNet50 indeed can match the performance of the dense WideResNet50, supporting the claims from our paper.
>
> We genuinely hope that you found the time to look over our response, in order to consider improving the paper rating, as you generously promised in your initial review. If there are still unclear aspects please let us know. Thank you!
>
> Best wishes,
>
> Authors

---

### Official Review · Reviewer_GL4s · 2021-11-03

**Correctness:** 4
**Technical Novelty And Significance:** 2
**Empirical Novelty And Significance:** 3
**Recommendation:** 6
**Confidence:** 4

**Main Review:**

Strengths:
1. The paper draws attention to the commonly used baseline method-random pruning and finds that it can be very effective for training a sparse network.
2. This paper explores and evaluates various random pruning methods with different layer-wise sparsity ratios. Adopting these layer-wise sparsity ratios as standard sparse initialization for future work seems necessary.
3. Experiments are promising, and analyses are very detailed.
4. The findings of the gradient norm seem consistent and correct, making good connections with previous works.
5. The paper also assesses random pruning from other perspectives for broader evaluation.

Weaknesses:
1. The effectiveness of random pruning has been investigated in previous works before, although not as detailed as in this paper.
2. There is not much novelty in the method itself. Mainly using multiple existing pruning methods to support their claim.

Questions for Author:
1. In the paper, you mentioned two categories (static and dynamic sparse training), and states that random pruning lies in the static sparse training. In the dynamic sparse training prune-and-grow scheme, if you randomly prune and grow the weights, can it be considered random pruning?
2. Can these observations and conclusions potentially apply to other popular models such as VGG or Transformer, etc.?
3. Reference styles are inconsistent, for example (Evci et al. , 2020a ) and Lee et al. (2018)

**Summary Of The Paper:**

This paper investigates the effectiveness of random pruning for sparse training. Specifically, the paper adopted several random pruning methods(ERK, Uniform, Uniform+) and compare the results with non-random Pruning methods(SNIP,GraSP). They are conducted on different ResNet scaling models, with  CIFAR-10/100 and ImageNet dataset under different  sparsity ratios. Many observations and conclusions were proposed in the experiments. Results show that random pruning can be quite effective for sparse training especially on larger or wider models.  In some cases, it can outperform well-versed pruning approaches and match the performance of dense networks.

**Summary Of The Review:**

This paper re-evaluates the underrated baseline of random pruning in sparse training. The paper is well-written and well-structured. Although the paper does not provide new pruning techniques,  I believe the conclusions from the paper can contribute to the sparse training domain.

---

> ### Author Response · Authors · 2021-11-19
> **Response to the reviewer GL4s**
>
> We sincerely appreciate your detailed comments and positive ranking. We address your questions below.
>
> **Q1: Novelty concerns**
>
> - Please see our overall response to all reviewers above for a full clarification of the novelty.
>
> **Q2: In the paper, you mentioned two categories (static and dynamic sparse training), and states that random pruning lies in the static sparse training. In the dynamic sparse training prune-and-grow scheme, if you randomly prune and grow the weights, can it be considered random pruning?**
> - Thank you! In this work, we mainly focus on pruning at initialization or static sparse training. We would rather not consider random pruning and regrowth in dynamic sparse training as random pruning due to its more complicated dynamical process in dynamic sparse training. For example, during training pruning even with random pruning serves as a regularization [1] along the training process, which can improve the generalization of the deep neural networks.
>
> **Q3: Can these observations and conclusions potentially apply to other popular models such as VGG or Transformer, etc.?**
>
> - Yes, we have added a set of experiments with VGG architectures in Appendix D in the revision. The results are in line with our original findings. The results are on par with our claim, i.e., the performance of training a randomly pruned sparse network quickly grows to match the dense network as the model size increases.
>
> **Q4: Reference styles are inconsistent, for example (Evci et al., 2020a ) and Lee et al. (2018)**
>
> Thank you for your question. We will try our best to align all the references in the next few days.
>
>
>
> [1] [Bartoldson, Brian R., et al. "The generalization-stability tradeoff in neural network pruning." arXiv preprint arXiv:1906.03728. NeurIPS 2020.](https://arxiv.org/abs/1906.03728)

---

> > ### Comment · Reviewer_GL4s · 2021-11-25
> > **Response to authors**
> >
> > Dear authors,
> >
> > Thank you for responding to my comments. The contributions as well as the concept of random pruning, are more clarified. The VGG results look positive.
> >
> > Best,
> >
> > Reviewer

---

### Author Response · Authors · 2021-11-19
**Response to all reviewers (2/2)**

## Suggested experiments


(1) **VGG (Reviewer GL4s).** -> Reviewer GL4s: Can these observations and conclusions potentially apply to other popular models such as VGG or Transformer, etc.?

- To validate if the observations and conclusions can generalize to other popular models, we conduct extra experiments with the VGG architecture on CIFAR 10/100. Due to the limited time and resources, we test ERK and SNIP ratios. We will continue to finish the experiments with other ratios. The results are updated as Appendix D in revision. The results are on par with our claim, i.e., the performance of training a randomly pruned sparse network quickly grows to match the dense network as the model size increases.

(2) **Comparison with dense Wide ResNet-50 (Reviewer Gr4w).**
-> Reviewer Gr4w: To claim random pruning is effective, what should be done is to compare randomly pruned WideResNet50 to a dense WideResNet50, not ResNet50. If the authors can provide this result and still show they have similar accuracy, I will improve my rating.
- We first explain that this comparison follows the convention of many prior works, e.g. [edgepopup] (https://openaccess.thecvf.com/content_CVPR_2020/papers/Ramanujan_Whats_Hidden_in_a_Randomly_Weighted_Neural_Network_CVPR_2020_paper.pdf) (who compared sparse Wide ResNet-50 with dense ResNet-34), among others. In this vein, we feel our way to compare (sparse Wide ResNet-50 > dense ResNet-50) will clearly show our “surprising” advances in a context.
- Second, we follow your advice to present the results of sparse versus dense Wide ResNet-50 in the following table. Indeed, we can even achieve comparable accuracies to Dense Wide ResNet-50 under a range of sparsities. Consequently, we humbly ask you to reconsider improving the rating of our paper, as you generously promised.
|  Methods  | | Wide ResNet-50  |ImageNet |  |
| ------------- | :-----------: |:-----------:|:-------------:|:-------------:|
| Sparsity | 0.85 | 0.70 | 0.60 | 0.50 |
| ERK+ | 70.27 | 75.20 | 76.70 | 77.43|
| SNIP | 73.48 | 76.81 | 77.55 | 78.10 |
| Dense | 78.11 | 78.11 | 78.11 | 78.11 |

(3) **Same scale of experiments on ImageNet (Reviewer WXbx).**
-> Reviewer WXbx: I still think this paper is a good asset and will be a great one if the same scale of experiments is done on ImageNet instead of CIFAR and will prove to be a solid exploratory work in that case.
- Thanks for your compliment and suggestions. We agree that the same scale of experiments on ImageNet is important. We have expanded the experiments on ImageNet with various architectures from ResNet-18 to ResNet-101 and Wide ResNet-50. Due to the limited time and resources, we choose the SNIP and ERK+sparsity ratios for Wide ResNet-50 and only SNIP ratios for the rest of the architectures. The sparsity of randomly pruned subnetworks is set as [0.7, 0.5, 0.3] from left to right.

- The results are shared in Appendix E. Overall, we observe a very similar pattern as illustrated in the original submission. On smaller models like ResNet-18 and ResNet-34, random pruning can not find matching subnetworks. When the model size gets considerably larger (ResNet-101 and Wide ResNet-50), random pruning can match the dense models at 50% sparsity.


(4) **FLOPs-Accuracy trade-off (Reviewer WXbx).**
-> Reviewer WXbx: While on the topic of sparsity ratios, I also noticed that the benchmarked pruning results don't have inference cost involved in these cases. For example, I would love to see the FLOPs for Fig 4 (right) to see what the FLOPs are for pruned WRN vs a Dense ResNet;
- We report the trade-off between accuracy and FLOPs of randomly pruned subnetworks and the corresponding dense networks in Appendix F. We can see that the number of FLOPs required by the pruned Wide ResNet-50 is still larger than the one required by the dense ResNet-50. Yet, the corresponding performance gain is also notable (2%). More importantly, when the model size is large enough, the accuracy of the randomly pruned sparse networks quickly grows and matches the corresponding large dense network (not only the small-dense models). In general, random pruning at initialization receives larger efficiency gains (the difference between the test FLOPs of sparse models and dense models) when the model size increases.

(5) **Broader Evaluation on ImageNet (Reviewer WXbx).**
-> I have no qualms about the broader evaluation except that I would love to see them on ImageNet even if it is for a single architecture. Is that something possible?
- As you suggested, we added the same evaluation on ImageNet as well in Appendix G. The performance on the broader evaluation on ImageNet is extremely similar to the test accuracy.  Random pruning can not match the performance of the dense network with small models. However, other important aspects, such as uncertainty and adversarial robustness benefit significantly with larger models.

---

### Author Response · Authors · 2021-11-19
**Response to all reviewers (1/2)**

We would like to thank the reviewers and the area chairs for their time and insightful comments. We are glad that they found our paper can be beneficial to our community. We provide a temporary revision to share the results of newly added experiments. We start by addressing some common comments:

&nbsp;
&nbsp;

## Novelty
While this paper does not provide any novel techniques, it does provide novel insights, large-scale experiments, and unique analysis.
**Our study is novel.**
- While prior works (Su et al, 2020; Frankle et al. 2020b) have observed that random pruning at initialization can be more competitive in certain cases, they did not give principled guidelines on when and how it can become that good; nor do they show it can match the performance of dense networks on ImageNet. Standing on the shoulders of giants, our work summarized principles by more rigorous studies, and demonstrate the strongest result so far, that randomly pruned sparse Wide ResNet-50 can be sparsely trained to outperform a dense Wide ResNet-50, on ImageNet.
Moreover, we show that ERK (Evci et al 2020) and our modified ERK+ are more general sparsity ratios that consistently demonstrate competitive performance without careful layer-wise sparsity design for every architecture, compared with the ad-hoc sparsity ratios used in Su et al, 2020. Specifically, ERK+ ratio achieves similar accuracy with the dense Wide ResNet-50 on ImageNet while being data free, feedforward free, and dense initialization free.

- Prior works on pruning of large models mainly focus on post-training (Wallace et al 2020) or during training (Gale et al, 2019). The performance of pruning, especially the random pruning, on large models in the promising field of (static) sparse training or pruning at initialization (PaI) is usually unknown. We believe the study of random pruning on PaI with large models is necessary for two reasons. (1) We are witnessing an era of model size explosion. The performance of the simplest random pruning at initialization on these large models is important for people who want to engage; (2) Frankle et al. 2020b showed that the behavior of PaI and post-training pruning is very different on moderate model sizes. However, the existing results of random pruning before training on large models are very scarce. To the best of our knowledge, the largest model evaluated on random pruning is ResNet-50 on ImageNet in Frankle et al. 2020b. In this paper, we show for the first time that random pruning performs very differently on large modes (e.g., ResNet-101, Wide ResNet-50) than smaller models and present compelling results that have been never explored before.

**Our findings are novel.**
- The performance of random pruning at initialization shown by our paper is unreasonable and novel. Before us, the performance of random pruning at initialization in literature is very limited and no prior works show that training randomly pruned networks can match the corresponding dense network with the same architecture on ImageNet. For instance, even though Frankle et al, 2020 show that randomly shuffled subnetworks can maintain or improve the original PaI methods,  their accuracy achieved by random pruning on ImageNet is quite unsatisfactory. According to the results of Figure 4 in Frankle et al, 2020, randomly pruned ResNet-50 at all sparsities even the trivial sparsity, i.e., 10%, 20%, can not match the accuracy achieved by the dense ResNet-50. This finding will significantly degrade the value of pruning at initialization or sparse training. On the contrary, our findings show that when the model size is large enough, training a 50% randomly pruned Wide ResNet-50 is sufficient to match the dense Wide ResNet-50.  We believe this finding is very important for the ML community. Never to mention, even the A100 GPUs supports 50% sparsity (true, just in a 2:4 structure). Just as the reviewer nAhU mentioned, “This is an inspiring new insight of broad potential impact in current ML trend, especially how to train larger models more efficiently.”

- The broader evaluation of random pruning is novel. We systematically evaluate random pruning from diverse aspects beyond test accuracy on both CIFAR and ImageNet (We have added a large range of experiments on ImageNet as suggested by the reviewer WXbx in the revision). The broader evaluation provides an important sanity check for random pruning at initialization in practice.

- We also add new insightful observations to the gradient flow literature, that only considering gradient flow at initialization might not be sufficient for sparse training. The gradient norm gap between sparse networks and dense networks could be more important.

---

### Decision · Program_Chairs · 2022-01-20

**Decision:**

Accept (Poster)

**Comment:**

### Summary

This paper builds on previous work on sparse training that shows the many modern sparse training techniques do no better than a random pruning technique that selects layer wise rations, but otherwise randomly selects which weights within a layer to remove.  The key difference in this work is to take these existing results and scale the size of the network to show that as the size of the network increases, the smaller -- as measured in pruning ratio -- a matching subnetwork becomes.

### Discussion

#### Strengths

Places an emphasis on simple techniques

#### Weakness

Significant overlap with previous work. Prior already demonstrated the equivalence of random pruning and contemporary pruning at initialization techniques.

### Recommendation

I recommend Accept (poster). However, I do want to stress that there is a significant overlap with previous work. The paper does appropriately attribute observations to previous work. However, there is some risk that readers may misinterpret the title and claim results as a wholly new observation about random pruning, where the reality is instead much more nuanced. Given that the work points to new methodological directions on considering scaling the network as an additional parameter to consider in pruning observations, I do believe these results -- even if narrower in scope that can be interpreted -- provide value to the community.